# Long-Term Evaluation of Intranigral Transplantation of Human iPSC-Derived Dopamine Neurons in a Parkinson’s Disease Mouse Model

**DOI:** 10.3390/cells11101596

**Published:** 2022-05-10

**Authors:** Sébastien Brot, Nabila Pyrenina Thamrin, Marie-Laure Bonnet, Maureen Francheteau, Maëlig Patrigeon, Laure Belnoue, Afsaneh Gaillard

**Affiliations:** 1Laboratoire de Neurosciences Expérimentales et Cliniques, Université de Poitiers, INSERM 1084, 86022 Poitiers, France; sebastien.brot@univ-poitiers.fr (S.B.); nabila.pyrenina.thamrin@univ-poitiers.fr (N.P.T.); marie.laure.bonnet@univ-poitiers.fr (M.-L.B.); maureen.francheteau@univ-poitiers.fr (M.F.); maelig.patrigeon@univ-poitiers.fr (M.P.); laure.belnoue@univ-poitiers.fr (L.B.); 2CHU Poitiers, 86022 Poitiers, France

**Keywords:** induced pluripotent stem cells, dopaminergic neurons, Parkinson’s disease, brain repair, recovery

## Abstract

Parkinson’s disease (PD) is a neurodegenerative disorder associated with loss of dopaminergic (DA) neurons in the substantia nigra pars compacta (SNpc). One strategy for treating PD is transplantation of DA neuroblasts. Significant advances have been made in generating midbrain DA neurons from human pluripotent stem cells. Before these cells can be routinely used in clinical trials, extensive preclinical safety studies are required. One of the main issues to be addressed is the long-term therapeutic effectiveness of these cells. In most transplantation studies using human cells, the maturation of DA neurons has been analyzed over a relatively short period not exceeding 6 months. In present study, we generated midbrain DA neurons from human induced pluripotent stem cells (hiPSCs) and grafted these neurons into the SNpc in an animal model of PD. Graft survival and maturation were analyzed from 1 to 12 months post-transplantation (mpt). We observed long-term survival and functionality of the grafted neurons. However, at 12 mpt, we observed a decrease in the proportion of SNpc DA neuron subtype compared with that at 6 mpt. In addition, at 12 mpt, grafts still contained immature neurons. Our results suggest that longer-term evaluation of the maturation of neurons derived from human stem cells is mandatory for the safe application of cell therapy for PD.

## 1. Introduction

PD is a neurodegenerative disorder associated with a progressive loss of DA neurons in the SNpc. Multiple therapeutic approaches have been developed to treat the symptoms of PD, such as pharmacological treatments with L-DOPA [1] or deep brain stimulation [2]. However, none of these approaches can stop or reverse the progression of DA neuron degeneration in the SNpc. An alternative strategy for treating PD is the transplantation of DA neurons to replace lost DA neurons. Indeed, because of the local and specific degeneration of mesencephalic dopaminergic (mDA) neurons, PD is an ideal candidate for cell therapy [3,4,5,6].

Cell replacement therapies in PD consist of restoring dopamine levels in the striatum by transplanting DA neuroblasts obtained from different sources, including fetal ventral midbrain (VM) tissue [7]. Experiments performed in animal models of PD have provided proof of principle that fetal VM tissue grafted into the striatum can survive, reinnervate the striatum, and improve motor deficits induced by nigrostriatal lesions. Clinical trials in patients with PD using intrastriatal transplantation of human fetal VM tissue have demonstrated that the transplanted neurons can survive for a long time, reinnervate the denervated striatum, and, in some patients, induce major clinical benefits [8,9,10,11]. The use of fetal tissue as a source of cells for cell therapies raises several issues such as a lack of cell standardization and significant ethical and logistical concerns [12,13]. Currently, other human cell sources, including human embryonic and hiPSCs, are under investigation for transplantation. Several studies have shown that transplantation of hiPSC-derived DA precursors restores dopamine levels in the striatum and promotes functional motor recovery in rodent [14,15,16,17] and nonhuman primate [18,19,20,21] models of PD. However, prior to the clinical application of DA neurons derived from hiPSCs, rigorous preclinical assessment of these cells should be conducted.

One of the crucial issues is to assess the long-term survival of these human cells after transplantation. Indeed, neuronal development and maturation in humans is a long and slow process. In most transplantation studies using human cells, the maturation of DA neurons has been analyzed over a relatively short period ranging from a few weeks to a few months, not exceeding 6 months [16,18,22,23,24,25,26]. In present study, we reprogrammed human fibroblasts into hiPSCs and differentiated these cells into mDA neurons. To evaluate the potential of these DA neurons to send long-distance axonal projections and to restore the nigrostriatal pathway, we grafted DA neurons homotopically into the SNpc. Graft survival and maturation were analyzed from 1 to 12 mpt. One of the key issues in the success of cell therapy applied to PD is the appropriate subtype of DA neurons used for transplantation. We analyzed the proportion of DA neurons of SNpc and ventral tegmental area (VTA) subtypes. At 6 mpt, we observed that the grafts contained nearly equal number of DA neurons of SNpc and VTA subtypes. However, by 12 mpt, we observed a decrease in the percentage of DA neurons of SNpc subtype, while the proportion of DA neurons of VTA subtype remained identical to that observed at 6 mpt. Moreover, we showed that even 12 months after transplantation, all grafted neurons were not mature; indeed, the graft still contained neuroblasts.

This was the first study showing the development and maturation of hiPSC-derived DA neurons grafted into the SNpc in an animal model of PD 1 year after transplantation. Our results show that an in-depth long-term evaluation of the development and maturation of neurons derived from human stem cells is mandatory for safe application of cell therapy not only for PD but for other brain diseases.

## 2. Materials and Methods

### 2.1. Fibroblast Reprogramming and hiPSC Culture

The hiPSC line used in this study was obtained by a reprogramming of human fibroblasts from healthy donor. Human dermis was recovered from surgical waste (abdominoplasty of a 28-year-old woman) at the University Hospital Center of Poitiers. On the same day as sampling, the biopsy was dissected and cultured to obtain fibroblasts. The fibroblasts were cultured on 6-well plates coated with 0.1% gelatin (Sigma-Aldrich, Saint Louis, MO, USA; G2500) in DMEM medium (ThermoFisher, Waltham, MA, USA; 31966-021) supplemented with 10% of fetal calf serum (ThermoFisher; 10500-67). The fibroblast reprogramming was carried out by scrupulously following the protocol of the StemMACS mRNA Reprogramming Kit (Miltenyi Biotec, Bergisch Gladbach Germany; 130-104-460). Briefly, fibroblasts were adapted to supplemented Pluriton Reprogramming Medium (Stemgent, Boston, MA, USA; 00-0070) for 2 passages before the cells were plated on CELL Start Substrate (LifeTechnologies, Carlsbad, CA, USA; A1014201)-coated plates on day -3 at a density of 4.0 × 10^4^ cells/well of a 6-well plate. On day 0, 100 µL of mRNA transfection complex (with a molar ratio of 3:1:1:1:1:1:1 of Oct4–Sox2–Klf4–Lin28–c-Myc–Nanog–nuclear eGFP, respectively) was added dropwise to the preplated cells. After 4 h, media change was performed with equilibrated, freshly supplemented Pluriton Reprogramming Medium plus B18R Recombinant Protein Carrier-free (0.5 mg/mL, eBiosciences, San Diego, CA, USA; 34-8185-81). These same steps were repeated for 11 consecutive days. On days 12–14, colonies were allowed to grow without further transfection, with a change of medium without B18R. On day 15, the colonies were observed under a microscope, picked manually with a micropipette, and dissociated with passaging solution XF (Miltenyi Biotec, 130-104-688). Then, the cells were cultured in feeder-free conditions in chemically defined IPS-Brew XF medium (Miltenyi Biotec; 130-104-368) supplemented with ROCK inhibitor Y27632 (10 µM, Miltenyi Biotec; 130-106-538) onto a 96-well plate coated with vitronectin (10 µL/mL, Life Technologies; A14700). Cells were incubated at 37 °C and 5% CO_2_, and the medium was changed every day. After reaching about 70% confluence, the hiPSCs were subcultured into 6-well plates to be expanded. At this stage, hiPSCs were cryopreserved in a mixture composed of 90% Knockout Serum Replacement (Gibco, Carlsbad, CA, USA; 10828-028) and 10% dimethyl sulfoxide (Sigma-Aldrich, D4540). A genetic stability assay was carried out by the company Stemgenomics (iCS-digital PSC 24-probes kit, Montpellier, France). The test relied on multiplex digital PCR with double-quenched probes and allowed targeting 24 genomic regions with efficient coverage of more than 90% of the most recurrent genomic defects described in hiPSCs.

### 2.2. Differentiation of hiPSC-Derived DA Neurons

The differentiation of hiPSCs into midbrain DA neurons was performed according to the protocol of the PSC Dopaminergic Neuron Differentiation Kit (Gibco; A3147701) with slight modifications. Briefly, 6-well plates coated with 10 μg/mL vitronectin were seeded with 4.0 × 10^5^ cells per well and incubated with complete IPS-Brew XF medium and 10 μM ROCK inhibitor Y27632 until reaching a confluence of 70%. The specification step was initiated by incubating the cells with 3 mL of complete Floor Plate (FP) Specification medium (Gibco; A31468-01) per well for 10 days. The specification medium was changed every 2 days. After 10 days, the culture consisted mainly of FP progenitor cells at 100% confluence. Then, the cells were subcultured three times to increase the pool of FP progenitor cells. StemPro Accutase (Gibco; A11105-01) was used for dissociation, and the cells were seeded with 3 mL of FP Cell Expansion medium (Gibco; A3165801) and 5 μM ROCK inhibitor Y27632 per well coated with laminin (10 μg/mL, Gibco; 23017-015). The medium was changed every 2 days without ROCK inhibitor during the 10 days of expansion stage. The original protocol recommended sphere formation for the third passage, but we obtained better results by passaging adherently one additional time to FP progenitor cells and then directly plating in maturation medium. FP progenitor cells were dissociated and seeded on a poly-D-lysine (100 μg/mL, Sigma-Aldrich; P7280) and laminin (15 μg/mL, Gibco; 23017-015) double-coated 24-well plate for 4.0 × 10^5^ cells per well in 500 μL of complete maturation medium (Gibco; A31474-01). The remaining cells were frozen with 90% expansion medium and 10% DMSO (Sigma-Aldrich; D4540) for 5.0 × 10^6^ cells each vial and were kept for further experiments. The seeded maturation cells were incubated at 37 °C and 5% CO_2_, and half medium was changed every 2–3 days before cells were fixed or used for transplantation.

### 2.3. Animals

All animal experimental procedures and housing were carried out in accordance with the European Communities Council Directive (2010/63/EU). All efforts were made to reduce the number of animals used and their suffering. The procedures referenced under the file number APAFIS#2019060311226483v2 were approved by ethics committee N°84 COMETHEA Poitou-Charentes. All mice were housed under the standard 12 h light/dark cycle. Animals had ad libitum access to food and water. We used a total of 55 RAG2 KO mice (B6.Cg-Rag2tm1.1Cgn/J from Jackson Laboratory, Bar Harbor, ME, USA) with a mix of males/females, of which 12 were used only as lesioned controls and 43 were lesioned and transplanted. All mice in the lesion group were kept for the duration of the experiment, i.e., approximately 1 year. For time course study, 7 mice were sacrificed at 1 mpt, 12 at 6 mpt, 11 at 8 mpt, and 13 at 12 mpt.

### 2.4. Lesion and Transplantation Procedures

Adult (4–6 months old) mice were lesioned by the injection of 6-hydroxydopamine (6-OHDA) into the SNpc as previously described [27]. Briefly, animals were anesthetized with a mixture of ketamine–xylazine (intraperitoneal, 100 mg/kg and 10 mg/kg of body weight, respectively). Animal received into the left SNpc 1 μL of 6-OHDA (8 μg/μL, Sigma-Aldrich; H116) dissolved in 0.9% saline containing 0.01% ascorbic acid (Sigma-Aldrich, A5960) solution using a 5 µL Hamilton syringe at the following stereotaxic coordinates: AP = −3.2 mm, ML = 1.4 mm, DV = −3.8 mm. The DA precursors used for transplantation were harvested on the 26th day of the differentiation protocol, i.e., after 5 days of neuronal maturation. For this, the cells were incubated at 37 °C for 3 min with accutase, then gently harvested without any trituration. The cells were centrifuged for 3 min at 300× *g* and resuspended at a concentration of at 100,000 cells/μL in maturation medium. Three weeks after the lesion, 1 µL of cell suspension (100,000 cells/µL) was injected into the SNpc at the same coordinates used for 6-OHDA injection.

### 2.5. Amphetamine-Induced Rotation

Rotational bias after amphetamine (4 mg/kg, intraperitoneally) was recorded using an automated system (Omnitech Electronics, Colombus, OH, USA). The animals were recorded for 60 min, and full body turns were counted and then expressed as net turns per minute. Data were expressed as net ipsilateral rotations (total left − total right 360° turns) per min. After postmortem histological analysis, only the mice exhibiting more than 65% of SNpc DA lost were used for behavioral analysis. Of the 25 mice used in this test, 1 mouse out of the 12 in the lesioned group and 2 mice out of the 13 in the grafted group exhibiting DA lesions of less than 65% were excluded from the behavioral analysis.

### 2.6. Tissue Processing

Mice were injected with a lethal dose of a mixture of ketamine–xylazine and perfused transcardiacally with 100 mL of saline (0.9%) followed by 200 mL of ice-cold 4% paraformaldehyde (PFA) in 0.1 M phosphate buffer (PB, pH 7.4). Brains were removed, postfixed in 4% PFA overnight at 4 °C, and cryoprotected in 30% (*w*/*v*) sucrose 0.1 M sodium phosphate solution (pH 7.4). For immunofluorescence experiments, brains were cut in 6 series on a freezing microtome (Microm HM450, ThermoFisher) in 40 µm-thick coronal sections and stored in a cryoprotective solution (20% glucose, 40% ethylene glycol, 0.025% sodium azide, 0.05 M phosphate buffer, pH 7.4).

### 2.7. Immunochemistry

#### 2.7.1. Immunofluorescence on Cells

The cells were fixed in 4% EM grade PFA (Electron Microscopy Sciences, Hatfield, PA, USA; RT-15710) for 15 min at room temperature (RT) at the different time points of observation. After washes with PBS 0.1 M, pH 7.4, the cells were permeabilized and blocked with 0.3% Triton X-100 and 3% BSA in PBS for 1 h at RT and incubated with primary antibodies diluted in blocking solution overnight at 4 °C. The next day, the cells were incubated with appropriate secondary antibodies diluted in blocking solution for 1 h at RT. Then, the cells were mounted with 15 µL of Vectashield Vibrance + DAPI (Vector Laboratories, Burlingame, CA, USA; H-1800-10) onto microscope slides for further observation.

#### 2.7.2. Immunohistochemistry on Brain Sections

Free-floating sections were incubated in a blocking solution (3% bovine serum, 0.3% Triton X-100 in PBS 0.1 M, pH 7.4) for 90 min at RT and then overnight at 4 °C in primary antibody solution. After washes with PBS 0.1 M, pH 7.4, sections were incubated in appropriate secondary antibodies diluted in blocking solution for 90 min at RT. Nuclei were labeled with DAPI (1:2000, Sigma-Aldrich; D9542). Sections were rinsed and mounted on slides and then coverslipped with DePeX (Sigma-Aldrich; 06522). A complete list of suppliers and concentrations of primary and secondary antibodies used is detailed in Appendix A. For DAB immunostaining, after primary antibody incubation (anti-hNCAM, 1:200, Santa Cruz Biotechnology, Santa Cruz, CA, USA; SC-106), sections were incubated at RT for 90 min with biotinylated secondary antibody (goat anti-mouse, 1:200, Vector Laboratories; BA-9200). To quench the endogenous peroxidases, sections were washed with Tris-HCl buffer (0.01 M pH 7.6) and incubated with 0.3% hydrogen peroxide (Sigma; 216763) for 30 min. Then, sections were incubated with an avidin–biotin peroxidase complex (Vectastain ABC Kit, Vector Laboratories; PK-61000) at RT for 60 min. The sections were subsequently rinsed in Tris-HCl buffer and then incubated with DAB/nickel substrate working solution (DAB peroxidase substrate kit, Vector Laboratories; SK-4100). The reaction was stopped using 0.1 M PBS when visible black staining was detected on the tissues. Slides were then dehydrated and covered with DePeX mounting medium.

### 2.8. Grafted Brain Clearing

A whole mouse brain after 12 mpt was used for iDISCO protocol as previously described [28]. The brain was cut in two in the sagittal plane and processed separately. Briefly, samples were dehydrated with gradual addition of methanol in distilled water (20%, 40%, 60%, 80%, 100% × 2, each for 1 h). Once dehydrated, samples were incubated overnight in a solution containing 66% dichloromethane (DCM, Sigma-Aldrich; 270997) in methanol (Alfa Aesar, Kandel, Germany; 31721) and then washed twice in 100% methanol. Overnight bleaching with a 1:5 ratio of hydrogen peroxide–methanol was performed at 4 °C. Tissues were then gradually rehydrated in PBS by removing methanol in 20% increments (1 h for each step). Detergent washing was then performed in PBS with 0.2% Triton X-100 (2 × 1 h). Pretreated samples were then incubated in PBS, 0.2% Triton X-100, 20% DMSO, and 0.3 M glycine at 37 °C for 2 days, then blocked in PBS, 0.2% Triton X-100, 10% DMSO, and 6% donkey serum at 37 °C for 2 days. The tissues were then incubated in primary antibody dilutions (anti-hNCAM, 1:200 and anti-TH, 1:500) in PBS-Tween 0.2% with heparin 10 µg/mL, 5% DMSO, and 3% donkey serum at 37 °C for 10 days. Samples were then washed in PBS-Tween 0.2% with heparin 10 µg/mL for 24 h (6 changes of the solution over the day) and incubated in secondary antibody dilutions (donkey anti-rabbit Alexa568, 1:500 and donkey anti-mouse Alexa647, 1:500) in PBS-Tween 0.2% with heparin 10 µg/mL, 5% DMSO, and 3% donkey serum at 37 °C for 7 days. Samples were washed with PBS-Tween 0.2% with Heparin 10 µg/mL for 24 h. For clearing, immunolabeled hemispheres were dehydrated with gradual addition of methanol in double-distilled water (20%, 40%, 60%; 80%, 100% × 2, each for 1 h) and incubated overnight in 33% methanol/66% DCM at RT. The methanol was then washed for 20 min twice in 100% DCM. Finally, tissues were incubated in ethyl cinnamate (Eci, Sigma-Aldrich; 112372) until clear (about 1 h) and then stored in Eci at RT. The cleared samples were imaged on a light sheet microscope (Ultramicroscope II; Miltenyi Biotec) with an Olympus MVX-10 Zoom Body equipped with a sCMOS camera (Andor Neo) and a 2×/0.5 objective lens (MVPLAPO 2×) equipped with a 6 mm working distance dipping cap. The samples were scanned with a step size of 3 μm using LED lasers (561 nm and 640 nm). Imaris software (Bitplane, Oxford Instruments, Belfast, United Kingdom) was used for the 3D reconstruction.

### 2.9. Data Acquisition and Quantification

For the quantification of cell cultures and grafted cells, images were acquired with a Zeiss Axio Imager.M2 Apotome microscope (Carl Zeiss, Oberkochen, Germany) at ×20 magnification. For cell cultures, at least 6 images per immunostaining and per experiment were taken. For nuclear stainings, cells were counted automatically using ImageJ software (Version 1.53c, NIH, Bethesda, MD, USA). For nonnuclear stainings, cells were counted manually. Areas of interest were further analyzed and photographed with a confocal laser-scanning microscope FV3000 (Olympus, Tokyo, Japan). The stereological analysis used was described previously [29]. Each section was scanned by a camera (Orca-R2, Hamamatsu Electronic, Hamamatsu City, Japan) connected to a microscope (DM 5500, Leica Microsystems, Wetzlar, Germany). Then, virtual sphere probes were scanned on the *Z*-axis of striatum or nucleus accumbens using the Mercator software (Explora Nova, La Rochelle, France). Each sphere was 8 μm in radius and contained in a 20 μm × 20 μm square; spacing between each square was 200 μm × 200 μm. Spheres were visualized as a series of concentric circles of changing circumferences upon focusing through the tissue. Finally, the intersections between the outline boundary of the sphere and the fibers were counted at each focal plane. The hNCAM fiber density was the number of fibers crossing the spheres along *Z*-axis.

### 2.10. Statistical Analysis

Statistical analyses of data were performed using the GraphPad software (Prism version 8.4.3, San Diego, CA, USA). For in vitro culture quantifications, the nonparametric Kruskal-Wallis test was used, and values are expressed as the mean ±SEM. Data are representative of at least 3 independent experiments. For quantifications of grafted cells on brain sections, the results are shown in the form of a violin plot with median and quartiles to show the distribution of the results among the transplants. For the quantification of fiber densities, values are the mean ± SEM with the distribution of individuals. The data on brain sections were analyzed with the nonparametric Kruskal-Wallis test, except for the comparison of DA subtypes between 6 and 12 mpt, for which a Mann-Whitney test was used. For amphetamine rotation tests, values are the mean ± SEM with the distribution of individuals, and the analysis of the various groups was performed with an RM two-way ANOVA with Geisser–Greenhouse correction followed by a Bonferroni’s multiple comparisons test. Correlations were computed with Pearson correlation coefficients. Differences were considered statistically significant when *p* < 0.05 (*), *p* < 0.01 (**), *p* < 0.001 (***).

## 3. Results

### 3.1. Generation of hiPSCs from Human Fibroblasts via mRNA Reprogramming

Currently, hiPSCs are widely considered the most suitable source of cells for autologous cell transplantation in PD. Nevertheless, the complexity and the costs of this approach make it unlikely for future clinical application. Current research has focused more on creating hiPSC banks in order to perform HLA-matched cell transplantation [30]. One of the crucial issues for clinical use of hiPSCs is the risk of mutagenesis and the integration of exogenous DNA into cellular genome. In this study, we used an mRNA reprogramming method that completely avoided the risk of genomic integration and insertional mutagenesis inherent to all DNA-based methods, including those that are ostensibly nonintegrating [31]. The mRNAs were complexed with a cationic vehicle, facilitating mRNA uptake by the fibroblasts, and the presence of GFP mRNA allowed monitoring the transfection efficiency over time (Figure 1A,B). After two weeks, we obtained a significant number of hiPSC colonies, approximately a dozen per 6-well plate well (Figure 1C). The colonies were manually picked and cultured under xeno-free conditions, i.e., without feeder cells and using a recombinant protein, the vitronectin, as attachment molecule (Figure 1D). The pluripotency of the hiPSC line was assessed by immunofluorescence, using several conventional pluripotency markers (Figure 1E). The hiPSCs highly expressed the markers SSEA-4 (96.2 ± 1.6%), OCT-4 (92.8 ± 1.7%), NANOG (94.5 ± 0.7%), LIN28 (87.1 ± 2.3%), and SOX2 (97.9 ± 0.5%) but did not express the differentiation marker SSEA-1(1.1 ± 0.3%) (Figure 1F). For all experiments, the hiPSC line used was kept below six passages. Furthermore, cell cultures were maintained until the 15th passage to check their stability and genetic integrity (Figure 1G). Digital PCR showed a consistent number of chromosomes and no genetic abnormalities in hiPSCs. These data showed efficient generation of a hiPSC line under xeno-free conditions using an mRNA reprogramming method.

### 3.2. hiPSC Differentiation to Midbrain Floor Plate Progenitor Cells

Several research groups have developed differentiation protocols to obtain DA neurons from human stem cells [32,33,34], and several commercial products for this purpose are currently available. We used a differentiation kit, which allowed obtaining highly reproducible results with a three-step protocol (Figure 2A). This protocol was similar to that developed by Parmar’s team, which demonstrated the possibility of obtaining a high yield of VM dopamine progenitors with a GMP grade [35]. In a 10-day first step, we differentiated hiPSCs into floor plate (FP) progenitor cells. These neural stem cells were multipotent and could therefore be amplified by successive passages or else be frozen for later use. After the expansion step, we checked in vitro the expression of the FP markers that are commonly used to confirm the mDA identity of progenitors (Figure 2B). At this stage, almost all the cells were positive for nestin, a marker widely used for neural stem cells. We quantified the number of cells positive for OTX2 (78.2 ± 3.2%), FOXA2 (77.1 ± 4.1%), LMX1A (80 ± 5.4%), and OTX2+/FOXA2+ (72.3 ± 3.6%) (Figure 2C). The optimal differentiation stage of DA neurons for transplantation depends in particular on the differentiation protocol used and can vary from day 16 to day 32 from one study to another [36]. In our case, we determined the optimal stage of DA neuron differentiation for transplantation, and we found the highest number of specific DA precursor markers at 5 days after the onset of neuronal maturation (day 26) (Figure 3A). At this stage, the cells had partially lost their proliferation capacity, as evidenced by a decrease in nestin expression (27.2 ± 0.9%; Figure 3B or Figure 4C). The expression of some FP markers, such as OTX2 (91.9 ± 2%) and FOXA2 (65.5 ± 2.4%), was maintained, while LMX1A (34.7 ± 1.5%) expression was decreased compared with day 21 (Figure 4C). In addition, 56.5 ± 1.4% of the cells expressed EN2, and 39.4 ± 2% of the cells expressed the DA precursor NURR1, a marker essential for early differentiation of midbrain DA neurons. Although the cells were collected at this stage for the transplantation experiments, we continued the in vitro neuronal maturation further to study the fate of the cells.

### 3.3. Maturation of DA Neurons In Vitro

To investigate the identity of hiPSC-derived cells, cultures were fixed at different stages of maturation and processed for immunofluorescence staining. At day 36 of differentiation, the FP markers FOXA2 (41.4 ± 1.7%) and LMX1A (35.9 ± 2.5%) continued to decrease, while the expression of NURR1 (35.4 ± 2.9%) was maintained (Figure 3C,D or Figure 4C). At this stage, only 5 ± 0.4% of the cells were positive for TH, the rate-limiting enzyme in the synthesis of DA neurons. The transcription factor PITX3, essential for the regulation of TH expression, was present in 24.9 ± 1.4% of the cells, and 11.2 ± 2.9% of the cells were positive for NeuN. These data indicate that hiPSCs were efficiently converted into a neural progenitor population corresponding to a mDA neural identity.

At day 51 of differentiation, 40.2 ± 2.6% of the cells were positive for NeuN, and 18.5 ± 4.7% were positive for TH (Figure 4A,B). The second type of neurons that we found was GABAergic neurons, with 5.1 ± 0.6% of cells expressing GAD67. Interestingly, we did not detect serotoninergic neurons in culture. It has been reported that the population of serotoninergic neurons present in the cells used for transplantation is partly responsible for dyskinesia, a major side effect observed in Parkinsonian patients grafted with VM fetal neural tissue [7,37]. We also observed 22% glial cells, among which 17.8 ± 2.5% were OLIG2+ and 4.5 ± 0.6% were GFAP+ (Figure 4B).

Next, we analyzed at various stage of differentiation the number of generated neurons positive for TH (Figure 4D). By day 26 of differentiation, only 4 ± 0.4% of neurons were TH+. The number of TH neurons gradually increased thereafter; at day 36, more than half of the generated neurons were TH+ (55.8 ± 7.2%; *p* = 0.018), and a proportion of 69.1 ± 5.1% (*p* = 0.0007) of TH+ neurons was reached at day 51. Not all DA neurons can be used for cell replacement therapy in PD [7]. The midbrain DA neurons are mainly divided into the A9 group of SNpc DA neurons, which undergo degeneration in PD, and the A10 group of VTA, which is largely unaffected in PD. Only the DA neurons of the SNpc subtype project to the dorsolateral striatum and regulate motor function. Therefore, one of the major goals in the field of cell therapy for PD is to generate, from stem cell sources, mDA neurons of nigral subtype.

Therefore, we determined the midbrain identity of TH+ neurons derived hiPSC by quantifying the number of TH+ neurons co expressing FOXA2 (Figure 4E,H). We found that 84.5 ± 2% of TH+ neurons co-expressed FOXA2, indicating the appropriate midbrain identity of the in vitro-generated TH+ neurons. Next, we analyzed the proportion of midbrain DA subtypes with the help of the commonly used markers GIRK2 and calbindin to distinguish DA neurons of the SNpc and VTA subtypes, respectively (Figure 4F,G). Our results showed the presence of two populations of DA neurons in culture. However, the proportion of the SNpc subtype was generally greater than that of the VTA subtype. Indeed, 58.8 ± 2.8% of TH+ neurons co-expressed GIRK2, and only 36.5 ± 1.8% of TH+ co-expressed calbindin (Figure 4I). Collectively, our findings indicated the reproducible and robust generation of large number of midbrain DA neuron from hiPSCs.

### 3.4. Maturation of hiPSC-Derived Neurons Takes Much Longer Than Previously Assumed

Neuronal development and maturation in human are a long and slow process. In most transplantation studies in PD using cells of human origin, the maturation of neurons has been analyzed over a relatively short period ranging from a few weeks to a few months [16,18,22,23,24,25,26]. In this study, graft survival and maturation were analyzed from 1 to 12 mpt (Figure 5A). After transplantation, to identify cells of human origin, we used anti-human neural cell adhesion molecule (hNCAM) and anti-human-specific nuclear (HuNu) antibodies. During transplantation, 100,000 cells were injected into the SNpc. In a first step, we investigated the survival of the grafted cells at different times post-transplantation (Figure 5B). The total number of grafted human cells was assessed by the HuNu marker. We did not observe significant differences in the number of cells from 1 to 12 mpt (*p* = 0.554). Indeed, we identified 28,817 ± 8407 HuNu+ cells at 1 mpt, 36,365 ± 3941 cells at 6 mpt, 43,994 ± 9007 cells at 8 mpt, and 46,767 ± 9495 cells at 12 mpt. These results are in agreement with a recent study showing an increase in the graft volume but not the number of cells over time [38].

We then focused on the neuronal maturation of grafted cells by quantifying immature and mature neurons expressing DCX and NeuN, respectively (Figure 5C and Figure 6A,B). We observed gradual maturation of the grafted neurons. At 1 mpt, 21.7 ± 1% of the grafted cells were NeuN+. At 6 mpt, 36.1 ± 2.4% were NeuN+. This was followed by a significant increase to 45 ± 2.4% at 8 mpt (*p* = 0.003) and 47.4 ± 2.4% at 12 mpt (*p* < 0.001). Surprisingly, at 12 mpt, we still observed the presence of immature neurons in the graft (Figure 6A,B). Indeed, we found that 17.7 ± 0.8% of HuNu+ cells co-expressed DCX. We also found 0.79 ± 0.5% of HuNu+ cells co-expressing DCX and NeuN. These results clearly showed that even one year after transplantation, more than a sixth of the transplanted neurons were still immature, and only half of the transplanted cells were mature neurons. Our results suggest that much longer survival times are needed to evaluate the development and maturation of neurons derived from human stem cells for safe application in cell therapy not only for PD but for other brain diseases.

Next, we investigated the population of DA neurons within the grafts by quantifying TH+ neurons from 1 to 12 mpt (Figure 5D). At 1 mpt, 18.8 ± 3.1% of the grafted neurons were TH+, and at 6 mpt, 23.9 ± 2.5% were TH+. We observed a significant increase in the number of TH+ neurons, reaching 34.9 ± 2.6% (*p* = 0.022) at 8 mpt and 33.7 ± 1.7% (*p* = 0.026) by 12 mpt. Evaluation of the percentage of TH+ neurons in the grafts showed a gradual increase in the proportion of DA neurons from 1 to 8 months and a stabilization at 12 months. We also evaluated the total number of TH+ cells in transplants from 6 to 12 mpt. We observed within the grafts 3627 ± 950 TH+ cells at 6 mpt, 5079 ± 1162 at 8 mpt, and 7689 ± 1621 at 12 mpt (Figure 5E). Since the proportion of DA neurons did not change between 8 and 12 mpt (Figure 5D), this result suggested that the neurons maturing beyond 8 mpt were mainly DA neurons.

### 3.5. Cellular Composition of the Graft

To characterize the final cellular composition of the grafts, the animals were sacrificed at 12 mpt to perform immunostainings (Figure 6, Figure 7 and Figure 8). We observed approximately 20–30% of HuNu negative cells within the graft region, suggesting their host origin. Next, we investigated the presence of glial cells in the graft (Figure 6A,C). We found that among HuNu positive cells, 14.7 ± 0.8% co-expressed GFAP+ and 23.6 ± 2.3% co-expressed OLIG2+. Furthermore, as observed in vitro, the second most present neuronal population after DA neurons was GABAergic neurons, with 12.6 ± 1.5% of neurons being GAD67+ (Figure 6A,D). We also investigated the presence of other types of neurons, such as serotonergic, cholinergic, and glutamatergic neurons, but none of these neuron types were found in the grafts.

One of the key issues in the success of cell replacement therapy applied to PD is the appropriate subtype of DA neurons used for transplantation. Accordingly, we investigated the midbrain identity of grafted DA neurons by quantifying the percentage of HuNu/TH/FOXA2+ cells (Figure 7A,B). These data showed that 67.1 ± 2.4% of TH+ cells had a midbrain identity. We analyzed the proportion of DA neurons co-expressing GIRK2 or calbindin to distinguish DA neurons of the SNpc and VTA subtypes, respectively (Figure 7A,C). At 6 mpt, we observed 52.5 ± 3.8% TH/GIRK2+ neurons and 39.9 ± 2.2% TH/calbindin+ neurons (Figure 7C). We found that the grafts contained nearly equal numbers of DA neurons of the SNpc and VTA subtypes. These results were in line with those of previous studies using human embryonic or induced pluripotent stem cell-derived DA neurons [25,39,40,41,42]. However, at 12 mpt, we observed a decrease in the percentage of TH/GIRK2+ neurons (*p* = 0.0013), while the percentage of TH/calbindin+ neurons remained identical to that observed at 6 mpt. Indeed, we observed 32.4 ± 2% TH/GIRK2+ neurons and 41.8 ± 1.4% of TH/calbindin+ neurons (Figure 7C). The decrease in the percentage of TH/GIRK2+ neurons observed at 12 mpt could be explained by the fact that the SNpc DA neurons were particularly vulnerable to cell death in comparison with other DA neurons such as those of the VTA subtype [43,44,45,46] or by the fact that the majority of TH neurons generated between 6 and 12 mpt were VTA-subtype DA neurons.

Once again, our results suggest the need for long-term studies to evaluate the therapeutic potential of DA neurons generated from stem cells. To our knowledge, this was the first study showing the development and maturation of hiPSC-derived DA neurons 1 year after transplantation in a mouse model of PD. Long-term studies are mandatory to evaluate in vivo the true impact of transplanted human neurons. In most transplantation studies, the survival of transplanted neurons has not exceeded 6 mpt. We showed the presence of a large number of DA neurons of the SNpc type at 6 mpt. However, between 6 and 12 mpt, while the number of TH+ neurons increased, the proportion of SNpc-subtype DA neurons decreased. Moreover, we showed for the first time that even 12 mpt, the grafts still contained neuroblasts.

### 3.6. Proliferation and Apoptosis in Grafts

Next, we investigated the proliferation and apoptosis in grafts. One of the critical issues after transplantation is the survival and proliferation rate of the transplanted neurons over time (Figure 8). Proliferation of human grafted cells was determined using Ki-67 in combination with HuNu markers (Figure 8A). First, we observed a relatively low rate of proliferating cells within the transplant. Second, we observed a significant decrease in the number of Ki-67 positive cells from 1 to 12 mpt (*p* = 0.0011): we found 9.78 ± 0.96% of Ki-67+ cells at 1 mpt, 1.56 ± 0.14% at 6 mpt, 0.27 ± 0.03% at 8 mpt, and 0.08 ± 0.03% at 12 mpt (Figure 8B). These results were in line with those of a previous study [15], showing that despite the fact that the cells were grafted at a highly proliferative stage, only 1.3% of proliferative cells were found at 6 mpt, and these cells all belonged to glial cell progenitors.

While it is well known that around 90% of cells die during or in the few days after transplantation [47], very few studies have investigated the death of grafted cells over a long period after transplantation. Here, we investigated the death of human grafted cells using cleaved capsase-3 (Casp-3) in combination with HuNu markers (Figure 8C). We observed a significant increase in the number of HuNu+/Casp-3+ cells from 1 to 12 mpt (*p* = 0.049): we identified 0.45 ± 0.13% of apoptotic cells at 1 mpt, 0.75 ± 0.14% at 6 mpt, 1.18 ± 0.37% at 8 mpt, and 1.35 ± 0.16% at 12 mpt (Figure 8D). Moreover, apoptosis seemed to affect more specifically DA neurons over time (Figure 8C,E). We observed a gradual increase in the number TH+/Casp-3+ cells within the graft over time. The percentage of TH/+Casp-3+ cells in the graft was 6.2 ± 2.8% at 1 mpt, 19.6 ± 2.6% at 6 mpt, 39.1 ± 4% at 8 mpt (*p* = 0.045), and 44.2 ± 4.3% at 12 mpt (*p* < 0.011). Since SNpc DA neurons are particularly vulnerable to cell death, this increase in apoptosis affecting TH+ cells could explain the decrease in the proportion of SNpc DA neurons that we observed after transplantation.

### 3.7. Reconstruction of the Nigrostriatal Pathway and Innervation of the Striatum

In most preclinical and clinical transplantation studies applied to PD, DA neurons are transplanted ectopically into the striatum, the target region of DA neurons, and not into the ontogenic region. To evaluate the potential of hiPSC-derived DA neurons to send long-distance axonal projections and to restore the nigrostriatal pathway, we grafted DA neurons homotopically into the SNpc [7,27].

To examine the progression and growth trajectories of axons of transplanted neurons, we performed a time course study following the sacrifice of transplanted animals at different mpt (1, 6, 8, and 12 months) (Figure 9). At 1 mpt, we observed hNCAM+ fibers leaving the transplant to the rostral direction, and a few graft-derived fibers were already present in the medial forebrain bundle (mfb). At 6 mpt, the number of hNCAM+ axons leaving the grafts increased, these fibers proceeding rostrally to the mfb and following the trajectory of the normal nigrostriatal pathway. These hNCAM+ axons terminated mainly in the CPu and the NAc. From 8 to 12 mpt, the density of hNCAM+ axons was increased in all regions examined, with the pattern of projections resembling that at 6 mpt. At 12 mpt, we observed a high density of hNCAM+ fibers in the CPu, including in the dorsolateral striatum (Figure 9A,B), the target of the A9 neurons of the SNpc, and in the NAc, the main target of the A10 neurons of the VTA (Figure 9E). Axon-derived grafted neurons were also observed along the medial forebrain bundle (Figure 9A,C,D), and the prelimbic cortex (PrL) (Figure 9F,F’).

We then quantified the axonal projections of the grafted neurons, using stereological analysis, within the CPu and NAc (Figure 9G,H). At 1 mpt, only a few hNCAM+ fibers were observed in the CPu. We observed 1.3 ± 0.7 at 6 mpt and a significant increase to 2.1 ± 0.4 at 8 mpt (*p* = 0.021). Interestingly, the density of the hNCAM+ axons doubled at 12 mpt (4.8 ± 0.7; *p* < 0.001) in comparison with 8 mpt (*p* = 0.07; Figure 9G). A similar innervation was also found for the NAc. We observed a progressive increased in the density of the hNCAM+ fibers at 6 mpt (3.4 ± 1.1) and at 8 mpt (7.9 ± 0.7; *p* < 0.001). However, unlike in the CPu, the hNCAM+ fiber density was stabilized at 12 mpt (5.1 ± 0.6; *p* = 0.028) in comparison with that at 8 mpt (Figure 9H). These results showed that the axons of the grafted neurons developed progressively, which was consistent with the progressive maturation of the grafted neurons.

Next, we used an original approach to investigate the 3D organization of the graft projections in the nigrostriatal pathway using clearing followed by the acquisition of an entire hemisphere of 12-month-grafted mouse with a light-sheet microscope. Graft-derived axonal projection was assessed using hNCAM antibody, allowing accurate labeling of the projections of grafted cells (Figure 10).

We observed a large number of fibers leaving the transplant and extending along the nigrostriatal pathway to the striatum (Figure 10A and Appendix A). At higher magnification, we observed that a large majority of hNCAM+ fibers co-expressed TH. We also observed a few fibers expressing only TH+, corresponding to the axons of the intact host DA neurons (Figure 10B). We found a large number of hNCAM+ fibers co-expressing TH in the striatum (Figure 10C,D and Appendix A). To confirm the mDA identity of graft-derived axonal outgrowth, we used a more specific DA marker the dopamine transporter (DAT). As DAT antibody is not compatible with the iDISCO protocol, we performed the labeling on brain sections. We found hNCAM/DAT+ neurons in the graft (Figure 10E). In the mfb, we also observed extensive graft axonal projections co-expressing hNCAM and DAT (Figure 10F). DA neurons grafted homotopically extend large numbers of axons over long distances to the striatum (Figure 11). Interestingly, the striatal areas innervated by axons of grafted neurons correspond to the regions normally innervated by DA neurons of SNpc. We next assessed the DA phenotype of striatal hNCAM+ fibers and found that nearly all hNCAM+ axons co-expressed TH (Figure 11A,C) or DAT (Figure 11B,D,E). We also observed a few axons expressing only TH or DAT, corresponding to the intact nigrostriatal projections. (Figure 10C,D).

In some cases, the transplants were not exactly placed at the level of the SNpc. Despite this, the grafted neurons were able to find their way to follow the nigrostriatal pathway and to reach their target region the striatum. This suggests that mDA neurons generated from hiPSCs are intrinsically specified, as we and others previously shown [25,48,49,50], to innervate the correct target structure. Taken together, in line with previous studies [22,27,32,51,52,53], we report that DA neurons derived from hiPSCs homotopically grafted into the lesioned SNpc sent extensive and specific DA projections to most targets of the host VM.

### 3.8. Long-Term Functional Motor Recovery after Transplantation

Three weeks after 6-OHDA injection and at 3, 6, and 10 months after transplantation, the grafted and lesioned animals were assessed for motor function using amphetamine-induced rotational behavior testing. First, we analyzed the extent of DA loss in the SNpc postmortem and motor deficits in 6-OHDA lesioned animals. The number of rotations induced by the amphetamine linearly correlated with the percentage of residual TH+ neurons in the SNpc (R = 0.72; *p* = 0.013) (Figure 12A). At 3 mpt, no significant difference (*p* = 0.178) was observed between the lesioned (3.14 ± 1.25 turns/min) and grafted groups (0.01 ± 0.73 turns/min). By 6 months postgrafting, the grafted animals showed a significant reduction (*p* = 0.02) in the number of rotations (0.09 ± 0.64 turns/min) compared with that of lesioned mice (3.03 ± 0.67 turns/min). At 10 mpt, grafted animals still showed a significant reduction in rotational behavior (*p* = 0.015); indeed, the amphetamine-induced rotations of the grafted animals were overcompensated (−1.25 ± 0.53) compared with those of lesioned animals (3.24 ± 1.17 turns/min) (Figure 12B). As expected, after the lesion, there was a positive correlation (R = 0.815; *p* = 0.026) between the rotation score and the percentage of lesion in the mice to be grafted. Interestingly, by 10 mpt, in the same group, this correlation was lost (R = 0.389; *p* = 0.388) (Figure 12C), suggesting that the DA neurons of the transplant replaced the neurons lost by the lesion.

A key factor in the therapeutic efficacy of transplanted DA neurons is the number of DA neurons to be used for transplantation. We have confirmed that, the functional recovery observed after transplantation is related to the number of DA neurons within the graft [54], with a negative correlation (R = −0.669; *p* = 0.049) and we determined that 2955 TH+ cells was necessary for functional improvement (Figure 12D).

## 4. Discussion

Our study focused on long-term evaluation of SNpc DA neurons derived from hiPSCs reprogrammed by mRNA transfection and grafted into the SNpc in a mouse model of PD. We reported that intranigral transplantation of SNpc DA neurons allowed the repair of the degenerated nigrostriatal pathway and induced functional recovery. However, we observed a decrease in the percentage of grafted SNpc DA neurons at 12 mpt. In line with this observation, we found a gradual increase in the number of apoptosis-affected DA neurons within the graft over time. Since SNpc DA neurons are particularly vulnerable to cell death, this increase in apoptosis affecting DA cells could explain the decrease in the proportion of SNpc DA neurons that we observed after transplantation.

Intrastriatal transplantation of human embryonic tissue obtained from VM in patients with PD has shown that grafted neurons can survive, reinnervate the striatum [55], release DA [56] and become functionally integrated in a patient’s brain. Although the results of these trials provided proof of principle for the cell replacement strategy in PD, there are several issues related to the sources of cells used for transplantation. Indeed, the use of tissue from human embryos poses ethical and logistical problems and a lack of standardization of cells intended for transplantation. Further studies are needed before this approach can be used as a routine treatment in patients. DA neurons obtained from human stem cells may address issues related to the use of fetal tissue for transplantation. In this study, we generated mDA neurons from hiPSCs, and these neurons expressed markers that normally expressed by mDA neurons during the development. These results were in line with those in previous reports [14,16,36,57,58,59,60].

One of the key factors that conditions the efficiency of the transplantation is the differentiation stage of the cells used for the transplantation. In a preliminary study, based on mDA marker expression, we selected the D25 and D26 differentiation stages for transplantation. As we did not find any difference between these two stages, we selected the D26 differentiation stage for transplantation. This stage of differentiation was in agreement with previous reports. For instance, it has been shown that day 25 of differentiation corresponds to the stage of high NURR1 expression, when most cells that have started to exit cell cycle (early postmitotic neurons) appear to produce rich mDA neuron grafts. This has resulted in successful rescue of complex motor abnormalities in animal PD models including mouse, rat, and monkey [23,61,62].

Several groups have focused on the identification of the most appropriate stage of development of DA neurons for transplantation. There has been high variability in the results reported. The optimal differentiation stage of differentiation DA neurons for transplantation has varied from day 16 to day 32 from one study to another [36]. For instance, Qiu et al. [63] compared three differentiated stages of cells generated by a FP-based protocol, DA progenitors (day 16), postmitotic immature DA neurons (day 25), and DA neurons (day 35), and examined the competence of these cells in cell-transplantation therapy in a mouse model of PD. They identified day 25 as the most suitable cell source for transplantation. Other groups transplanted the progenitors later, at day 28 mDA [20,40], or at relatively early stage, at day 16 mDA [22,34,39,53,64,65]. The conclusion of these studies was that the transplantation of the cells starting from the late midbrain floor-plate to an mDA neuroblast stage or early mDA neurons may be suitable for robust survival and function of grafted neurons.

An important element of evaluation of the suitability of transplanted cells for cell therapy in clinic is the assessment of long-term survival and neuroanatomical and functional integration of these cells into the host brain. Among the critical issues after transplantation are the survival, proliferation, and maturation of the transplanted neurons. In this study, we evaluated these three parameters on grafted neurons from 1 to 12 mpt. We observed a relatively low rate of proliferating cells within the transplant and a significant decrease in the number proliferating cells from 1 to 12 mpt. These results were in line with a previous study [57]. Only 1.3% of proliferative cells were found at 6 mpt, and these cells all belonged to glial cell progenitors.

While it is well known that around 90% of cells die during or in the few days after transplantation [48], very few studies have investigated the death of grafted cells over a long period after transplantation. Here, we investigated the death of grafted cells from 1 to 12 mpt. We observed a significant increase in the number of apoptotic cells from 1 to 12 mpt. Moreover, apoptosis seemed to affect more specifically DA neurons. Indeed, we observed a gradual increase in the number apoptotic cells affecting DA neurons within the graft over time. Since SNpc DA neurons are particularly vulnerable to cell death, this increase in apoptosis affecting TH+ cells could explain the decrease in the proportion of SNpc DA neurons that we observed after transplantation. To our knowledge, this was the first study showing death of DA neurons even 12 months after transplantation. This is an important point to consider in the clinical setting. We also report that neuronal maturation and axonal growth continued up to one year after transplantation. Indeed, we showed for the first time the presence of immature neurons within transplants up to 12 mpt. In most transplantation studies, the survival of transplanted neurons has not exceeded 6 mpt.

Another important point related to the transplantation of mDA neurons is the site of cell transplantation. In most preclinical and clinical transplantation studies applied to PD, DA neurons are transplanted ectopically into the striatum, the target region of DA neurons, and not into the ontogenic region. To evaluate the potential of hiPSC-derived DA neurons to send long-distance axonal projections and to restore the nigrostriatal pathway, we grafted DA neurons homotopically into the SNpc. We report that DA neurons derived from hiPSCs homotopically grafted into the lesioned SNpc sent extensive and specific DA projections to most targets of the host VM. These results were in line with those of previous studies using fetal mDA neurons [27,51,66] or mDA neurons generated from human pluripotent stem cells [22,39,53,67] after homotopic grafting into rodent models of PD.

Currently, several clinical trials of intrastriatal transplantation of DA neurons derived from hiPSCs are under evaluation in patients with Parkinson’s disease [68,69]. The first clinical and imaging results of these trials suggested possible benefits of cell transplantation over a period 24 months. Our present findings and previously published data [27,66] showed the beneficial effects of intranigral transplantation of DA neurons and suggest the possibility of considering intranigral transplantation in patients with Parkinson’s disease. However, several issues remain to be considered in clinic, such as the distance between the SNpc and the striatum, which is longer in humans than in rodent models of PD, and the extent and specificity of DA repair. Combinatorial approaches such as combined cell and biomaterial scaffolds [70] or a combined cell and gene therapy approach [71] are necessary to obtain more complete restoration of degenerated DA pathways. Currently, several innovative approaches are being tested to increase the efficiency of intranigral transplantation. For instance, Moriarty et al. reported that the overexpression of glial-cell-line-derived neurotrophic factor in the striatum of a rat model of PD stimulated axonal growth from grafted mDA neurons derived from hiPSCs and ameliorated motor function [71]. Interestingly, GDNF delivery may also have a neuroprotective effect on host DA neurons still present in the SNpc [72,73].

One limitation of our study is that only one hiPSC line was used to generate mDA neurons. Despite the fact that we obtained results very similar to those observed in the literature [14,36], the differentiation in DA neurons can vary from one cell line to another, which could impact the cellular composition and therefore the efficiency of the transplant. To date, studies have focused more on creating hiPSC banks in order to perform autologous transplants with HLA matching or on creating universal donor hiPSCs by genome editing using technologies such as CRISPR–cas9 [30]. Otherwise, other approaches provide clinical proof of concept for in vivo CRISPR–Cas9-mediated gene editing as a therapeutic strategy [74], especially for familial forms of PD.

## 5. Conclusions

In conclusion, we showed that midbrain DA neurons derived from hiPSCs obtained through mRNA reprogramming strategies, grafted homotopically into the 6-OHDA lesioned SNpc, sent axons to the targets of mDA neurons. We also report long-term motor functional recovery after transplantation, which was correlated to the number of DA neurons within the graft. Surprisingly, we found within the graft a decrease in the percentage of SNpc DA subtype from 6 to 12 mpt and the presence of immature neurons even 12 months after grafting. These observations have important implications for the design of human clinical trials. Accordingly, relying on short survival times could lead to misinterpretation of the results. The therapeutic evaluation of transplanted human cells will have to take into account the extended development time of human neurons.

Our results show that much longer survival times are needed to evaluate the development and maturation of neurons derived from human stem cells for safe application of cell therapy not only for PD but for other brain diseases.

## Figures and Tables

**Figure 1 cells-11-01596-f001:**
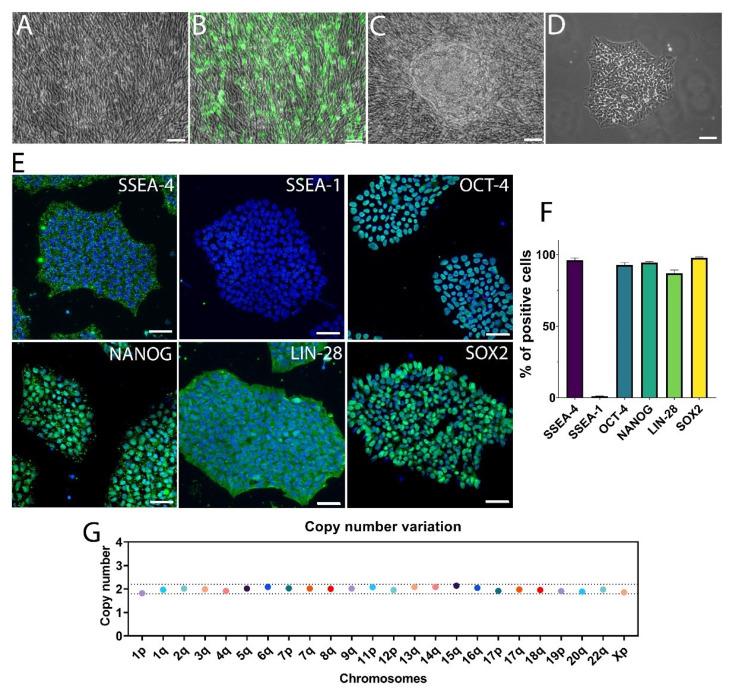
Reprogramming and pluripotency of hiPSCs. (**A**–**D**) Representative images in phase contrast of the reprogramming process of human fibroblasts into hiPSCs. During the first 12 days of the protocol, fibroblasts (**A**) were transfected daily with a cocktail of mRNA. (**B**) The transfection rate was checked with GFP nuclear expression (green). Around 12–14 DIV, colonies of hiPSCs (**C**) appeared within human fibroblasts and could be manually picked. (**D**) The subcultured hiPSCs were cultured under xeno-free conditions, in the absence of feeder cells. Scale bars: 50 μm. (**E**) Immunofluorescence images of hiPSCs expressing SSEA-4, SSEA-1, OCT-4, NANOG, LIN-28, and SOX2 markers (in green). The nuclei were labeled with DAPI (in blue). Scale bars: 50 μm. (**F**) Percentage of hiPSCs positive for different markers. A large majority of cells were positive for pluripotency markers SSEA-4, OCT-4, NANOG, LIN-28, and SOX-2, and negative for differentiation marker SSEA-1. Data are expressed as means ± SEM (*n* = 3 biologically independent experiments with ≥2000 cells per experiment). (**G**) The number of copies for each chromosome in the hiPSC line showed no abnormalities.

**Figure 2 cells-11-01596-f002:**
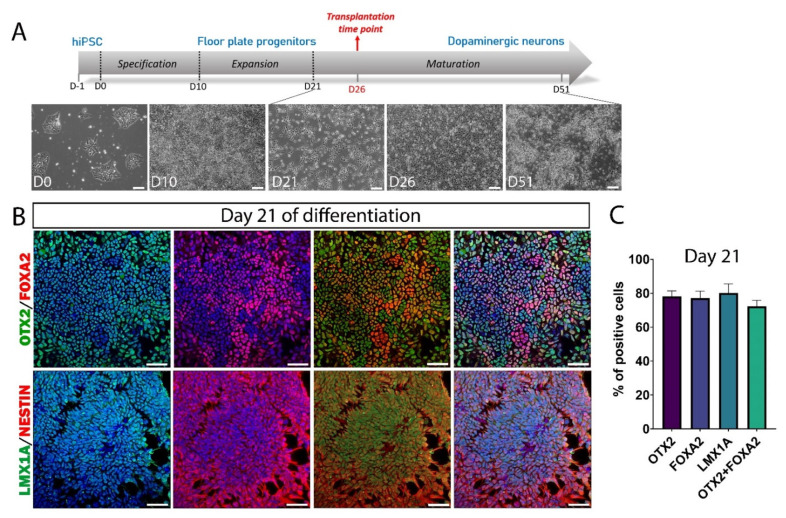
hiPSC induction into midbrain FP progenitor cells. (**A**) Schematic representation of DA neuron differentiation stages: specification, expansion, and neuronal maturation. Scale bars: 100 μm. (**B**) Representative pictures of the expansion step (day 21) showing the expression of OTX2, FOXA2, LMX1A, and nestin in FP progenitor cells. Merge images are shown with or without DAPI staining (in blue). Scale bars: 50 μm. (**C**) The quantitative analysis showed that almost 80% of cells expressed the FP progenitor markers and almost 70% of OTX2+ co-expressed FOXA2. Data are expressed as means ± SEM (*n* = 3 biologically independent experiments with ≥20,000 cells per experiment).

**Figure 3 cells-11-01596-f003:**
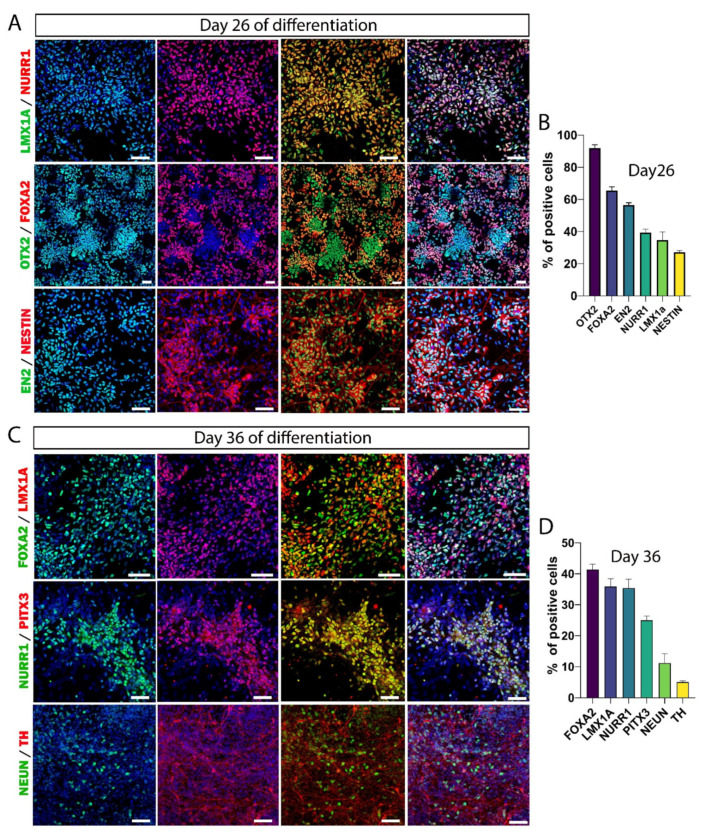
FP progenitor cell induction into DA precursors. (**A**) Representative immunofluorescence pictures of DA precursor marker expression at day 26. Merge images are shown with or without DAPI staining (in blue). Scale bars: 50 μm. (**B**) Quantitative analysis of different markers expressed by DA precursors at day 26. Approximately 30–40% of cells expressed markers of DA precursors, while the proportion of nestin+ cells was approximately 25%. (**C**) Representative immunofluorescence pictures of cells positive for the DA precursor markers, NeuN and TH at day 36. Merge images are shown with or without DAPI staining (in blue). Scale bars: 50 μm. (**D**) Quantitative analysis of different markers expressed by DA precursors at day 36. The percentage of cells expressing the later DA precursor marker PITX3 was around 25%. At this stage of maturation, more than 10% of the cells expressed NeuN, but only 5% of the NeuN+ cells co-expressed TH. Data are expressed as means ± SEM (*n* = 3 independent experiments with ≥8000 cells per experiment for day 26 and ≥12,000 cells per experiment for day 36).

**Figure 4 cells-11-01596-f004:**
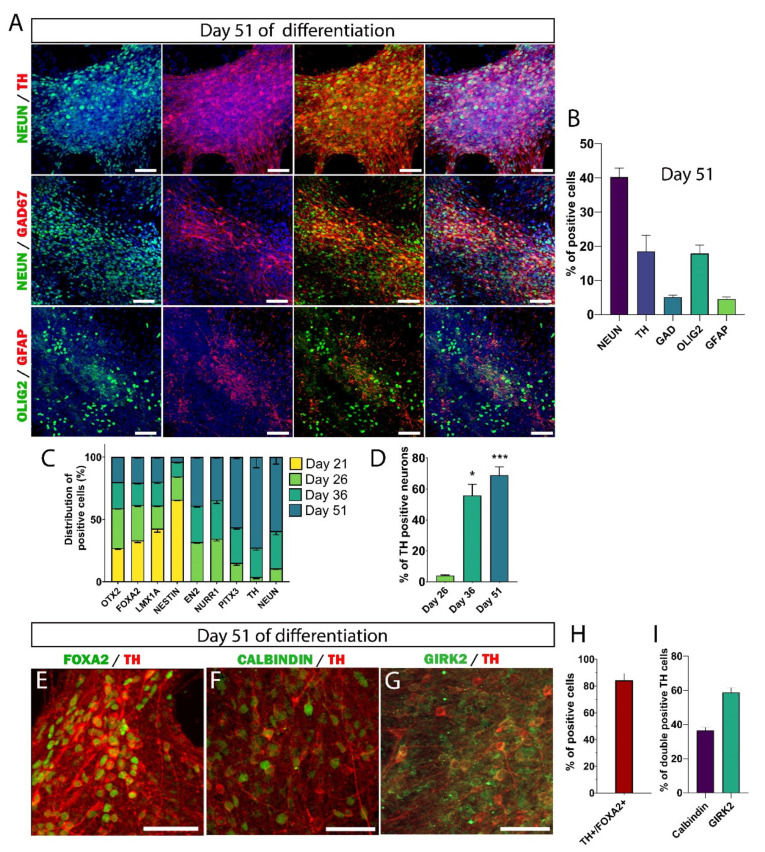
Characterization of the cell population after one month of maturation (day 51). (**A**) Immunofluorescence pictures of neurons and glial cells at day 51. Merge images are shown with or without DAPI staining (in blue). Scale bars: 50 μm. (**B**) Quantitative analysis of neurons and glial cells at day 51. Of the cells, ~40% were mature neurons expressing NeuN, 18% were oligodendrocytes, and less than 5% were astrocytes. Nearly 20% of the total cells expressed TH and 5% expressed GAD67. Data are expressed as means ± SEM (*n* = 3 biologically independent experiments with ≥17,000 cells per experiment). (**C**) Histograms showing the proportion of the cells expressing the markers of specification, differentiation, and maturation from day 21 to day 51. Data are expressed as the mean ±SEM (*n* = 3 independent experiments). Number of cells counted per experiment: ≥20,000 for day 21, ≥8000 for day 26, ≥12,000 for day 36, and ≥17,000 day 56). (**D**) Quantification of the number of TH+ neurons at different stages of maturation. At the beginning of maturation (day 26, time-point chosen to harvest the cells for transplantation), there were relatively few TH+ neurons (approximately 4%). The number of TH neurons increased gradually to reach 56% at day 36 and 70% at day 51. Data are expressed as means ± SEM. * *p* < 0.05 day 26 versus day 36, *** *p* < 0.01 day 26 versus day 51 (*n* = 3 biologically independent experiments with ≥12,000 cells per experiment). (**E**–**G**) Representative pictures of TH+ neurons co-expressing FOXA2, calbindin, or GIRK2 at day 51. Scale bars: 50 μm. (**H**) The quantification of subtype DA neuron markers showed 85% TH/FOXA2+ cells, confirming the midbrain identity of DA neurons. Data are expressed as means ± SEM (*n* = 3 independent experiments with ≥1300 cells per experiment). (**I**) Quantification of A9 and A10 DA neuron subtypes at day 51. Nearly 60% of TH+ cells were GIRK2+, while less than 40% were calbindin+. Data are expressed as means ± SEM (*n* = 3 biologically independent experiments with ≥1000 cells per experiment).

**Figure 5 cells-11-01596-f005:**
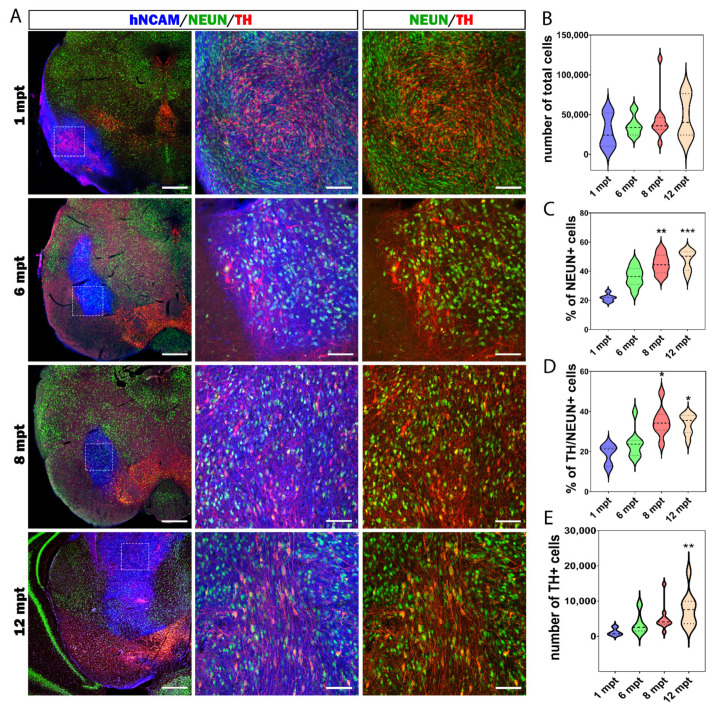
Maturation of neural stem cells grafts. (**A**) Representative images of NeuN+ and TH+ cells at 1, 6, 8, and 12 mpt within the graft as identified by hNCAM (in blue). Scale bars: 500 μm for the left column and 100 μm for the middle and right columns. (**B**) Quantification of the number of grafted cells from 1 to 12 mpt. The total number of the cells within the graft was stable from 1 to 12 mpt (*p* = 0.5547). (**C**) Quantification of NeuN+ cells within the graft from 1 to 12 mpt. The percentage of NeuN+ cells in the graft increased from ~20% at 1 mpt to reach ~50% at 12mpt. (**D**) Quantification of TH/NeuN+ neurons within the graft from 1 to 12 mpt. The percentage of TH+ neurons increased from 1 to 8 mpt and remained stable at around 35% at 12 mpt. (**E**) Quantification of the total number of TH+ neurons within the graft. From 1 to 12 mpt, the number of TH+ neurons within the graft increased to reach 7500 cells at 12 mpt. The results are shown in the form of violin plots with median and quartiles. * *p* < 0.05 1 mpt versus 8 and 12 mpt, ** *p* < 0.01 1 mpt versus 8 and 12 mpt, *** *p* < 0.001 1 mpt versus 12 mpt (*n* = 4 mice for 1 mpt, 10 mice for 6 mpt, 10 mice for 8 mpt, and 9 mice for 12 mpt).

**Figure 6 cells-11-01596-f006:**
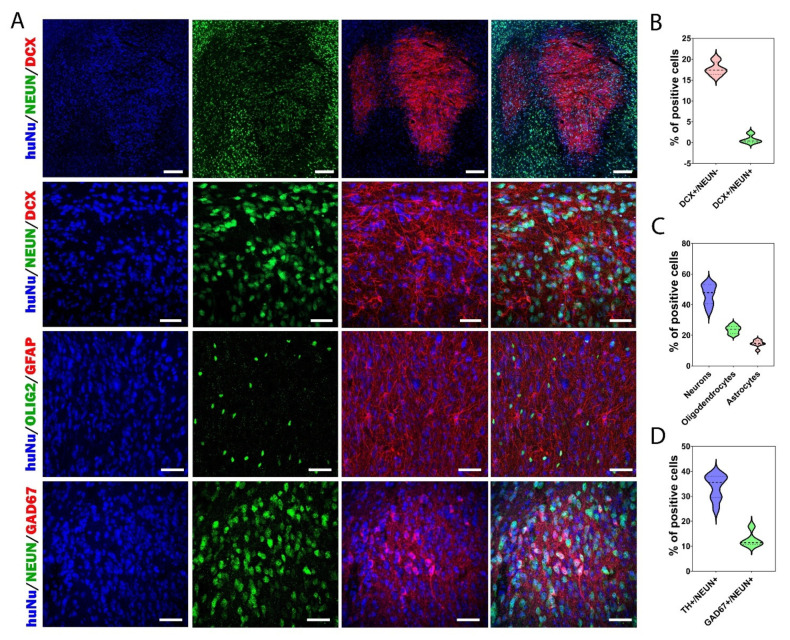
Cellular composition of the grafts at 12 mpt. (**A**) Representative images of different neuronal and glial markers in graft. Scale bars: 50 μm. (**B**) The proportion of immature neurons within the graft. The percentage of DCX+/NeuN- neurons was around 15%, while the proportion of DCX+/NeuN+ cells was very low. (**C**) Percentages of mature neurons, oligodendrocytes, and astrocytes within the graft. Half of the grafted cells were mature neurons, while ~20% were oligodendrocytes and ~15% were astrocytes. (**D**) Percentages of NeuN/TH+ and NeuN/GAD67+ neurons within the graft. About 35% of neurons were DA neurons, and around 12% of the mature neurons were GABAergic neurons. The results are shown in the form of a violin plot with median and quartiles (*n* = 9 mice).

**Figure 7 cells-11-01596-f007:**
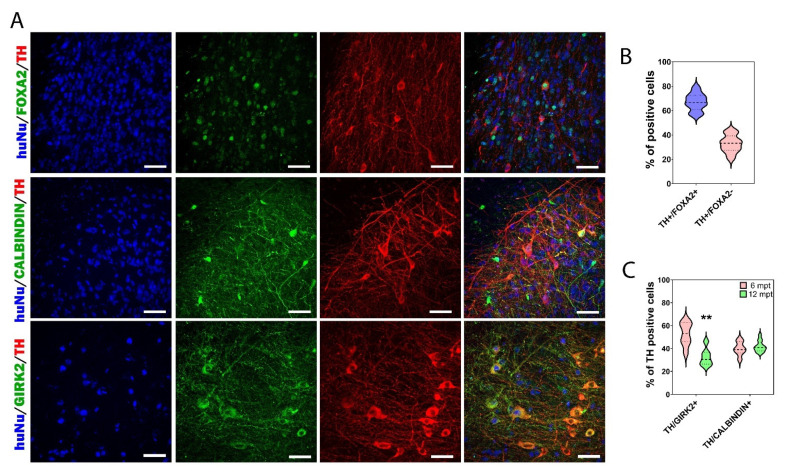
DA neuron subtypes within the grafts at 12 mpt. (**A**) Representative images of grafted HuNu+ cells expressing FOXA2, calbindin, or GIRK2 markers. Scale bars: 50 μm. (**B**) Percentages of TH/FOXA2+ and TH/FOXA2- within the graft; ~65% of grafted TH+ cells had midbrain identity. (**C**) Percentages of A9 and A10 DA neuron subtypes in graft at 6 and 12 mpt. At 6 mpt, 52% of TH+ neurons co-expressed GIRK2, and 40% of TH+ neurons co-expressed calbindin. At 12 mpt, a decrease of ~20% in TH/GIRK2+ was observed, while the percentage of TH/calbindin+ DA neurons remained stable. The results are shown in the form of a violin plot with median and quartiles. ** *p* < 0.01 6 mpt versus 12 mpt (*n* = 7 mice for 6 mpt and 9 mice for 12 mpt).

**Figure 8 cells-11-01596-f008:**
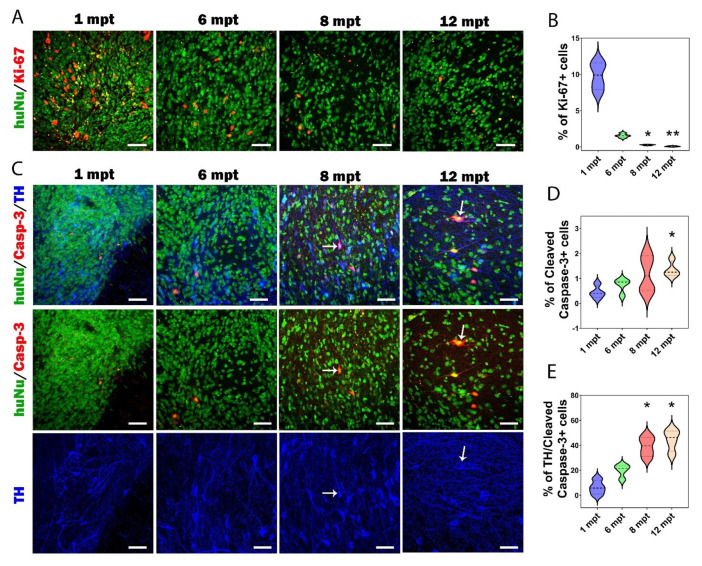
Cell proliferation and cell death in grafts over time. (**A**) Representative images of grafted HuNu+ cells (in green) expressing proliferative marker Ki-67 (in red) at 1, 6, 8, and 12 mpt. Scale bars: 50 μm. (**B**) Histogram showing the percentage of Ki-67+ cells within the graft from 1 to 12 mpt. (**C**) Representative images of cleaved caspase-3+ (in red) and TH+ cells (in blue) at 1, 6, 8, and 12 mpt within the graft as identified by HuNu (in green). White arrows indicate co-expression of HuNu, Casp-3, and TH markers. Scale bars: 50 μm. (**D**) Histogram showing the percentage of cleaved caspase-3+ cells within the graft from 1 to 12 mpt. (**E**) Histogram showing that the percentage of TH+ cells co-expressing cleaved caspase-3+ increased from 1 to 12 mpt. The data are shown in the form of a violin plot with median and quartiles. * *p* < 0.05 1 mpt versus 8 mpt and 12 mpt, ** *p* < 0.01 1 mpt versus 12 mpt (*n* = 4 mice for 1 mpt, 4 mice for 6 mpt, 4 mice for 8 mpt, and 4 mice for 12 mpt).

**Figure 9 cells-11-01596-f009:**
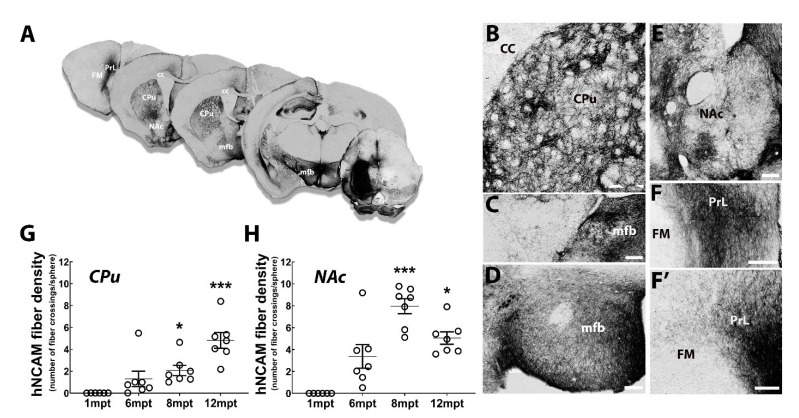
Time course of graft axonal outgrowth. (**A**) Illustrated overview of hNCAM+ fiber outgrowth from intranigral transplant at 12 mpt. (**B**–**F**’) High-magnification images showing the distribution of hNCAM+ fibers in the CPu (**B**), anterior (**C**) and posterior (**D**) mfb, NAc (**E**), and PrL (**F**,**F**’) at 12 mpt. Scale bars: 200 μm. (**G**,**H**) Quantification of hNCAM+ fiber density in the CPu (**G**) and NAc (**H**) at different timepoints post-transplantation. Means with the distribution of individuals and SEM are presented. * *p* < 0.05 1 mpt versus 8 mpt and 12 mpt, *** *p* < 0.001 1 mpt versus 8 mpt and 12 mpt (*n* = 6 mice for 1 mpt, 7 mice for 6 mpt, 7 mice for 8 mpt, and 7 mice for 12 mpt). Abbreviations: CC: corpus callosum, CPu: caudate putamen, FM: forceps minor, mfb: medial forebrain bundle, NAc: nucleus accumbens, PrL: prelimbic cortex.

**Figure 10 cells-11-01596-f010:**
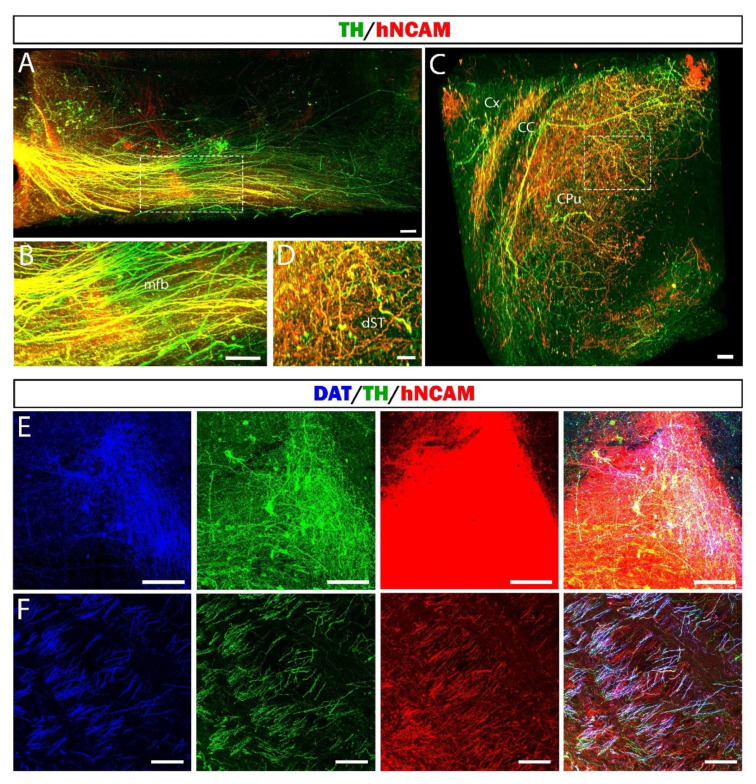
Three-dimensional visualization of whole mouse brain after 12 mpt. (**A**–**D**) Image representing a 3D reconstruction of the graft projections after the acquisition with a light sheet microscope of a mouse hemisphere cleared by the iDISCO technique at 12 mpt. (**A**) Representative image of the 3D visualization of the nigrostriatal pathway. Scale bar: 100 μm. (**B**) High magnification of dotted square in A showing hNCAM/TH+ fibers in the nigrostriatal pathway. Scale bar: 100 μm. (**C**) Representative image of the 3D visualization of the graft projections within the CPu. Scale bar: 150 μm. (**D**) High magnification of dotted square in C showing hNCAM/TH+ fibers in the dST. Scale bar: 100 μm. (**E**,**F**) Representative images of triple DAT/TH/hNCAM labeling in the graft (**E**) and along the nigrostriatal pathway (**F**) on brain sections at 12 mpt. Scale bars: 50 μm. CC: corpus callosum, CPu: caudate putamen, Cx: cortex, dST: dorsal striatum, mfb: medial forebrain bundle.

**Figure 11 cells-11-01596-f011:**
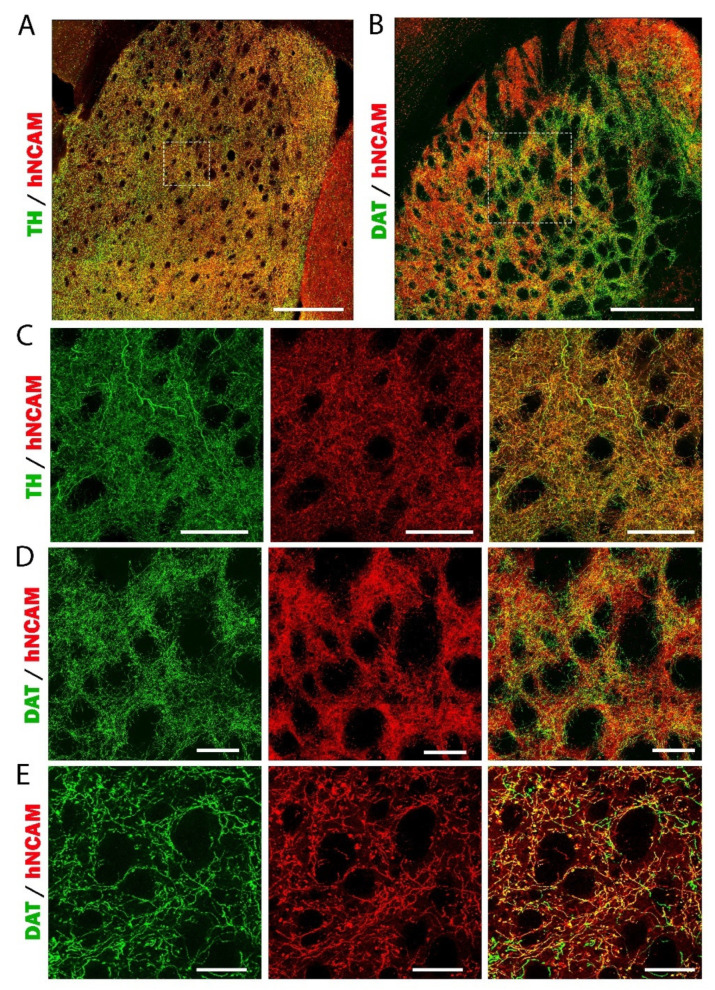
DA identity of hNCAM+ fibers in CPu. (**A**,**B**) Representative images of hNCAM/TH+ (**A**) and hNCAM/DAT+ axons (**B**) in the striatum. Scale bars: 500 μm. (**C**,**D**) High magnification of the dotted squares in A and B, respectively, showing the colocalization of hNCAM+/TH+ (**C**) and hNCAM+/DAT+ axons (**D**). Scale bars: 100 μm. (**E**) Higher-magnification images in the striatum showing large numbers of hNCAM+/DAT+ axons, confirming the DA identity of the graft projections to the striatum. Scale bars: 50 μm.

**Figure 12 cells-11-01596-f012:**
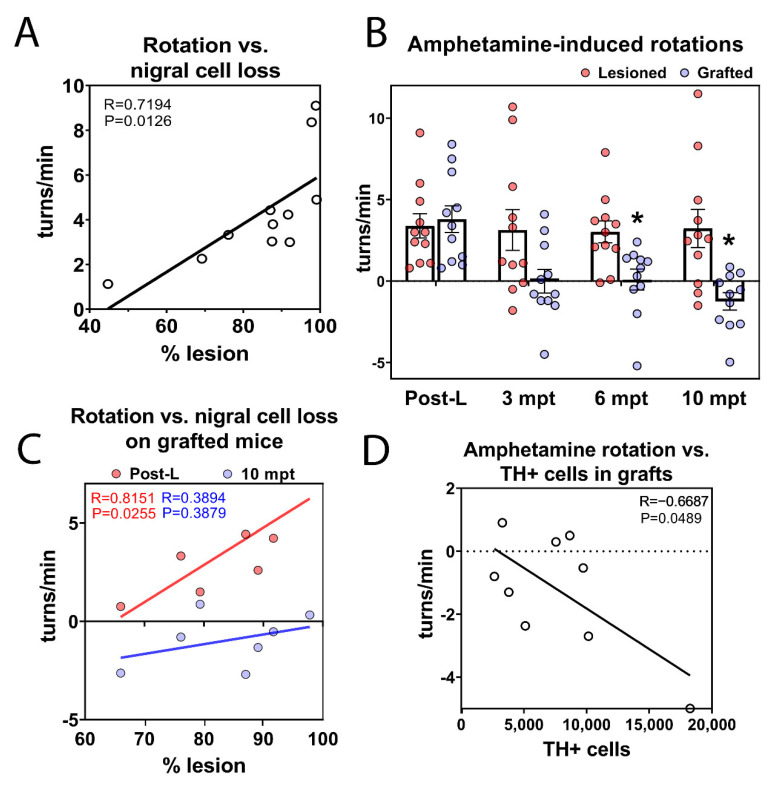
**Functional motor recovery after transplantation.** (**A**) Three weeks after 6-OHDA injection, correlation between the number of drug-induced rotations and the percentage of DA lesion. (**B**) Amphetamine-induced rotations in lesioned (in red) and grafted (in blue) mice at three weeks after the lesion (Post-L) and 3, 6, and 10 mpt. Functional recovery in the grafted animals was observed at 6 mpt compared with the lesioned mice, and this effect lasted up to 10 mpt. Mean with the distribution of individuals and SEM are presented. * *p* < 0.05 lesioned versus grafted (*n* = 11 lesioned mice and 11 grafted mice). (**C**) Correlation between the number of drug-induced rotations and the percentage of DA lesion in grafted mice at Post-L (in red) or at 10 mpt (in blue). The results showed a significant correlation between the two parameters after lesion but not at 10 mpt. (**D**) Correlation between the number of drug-induced rotations and the number of TH+ neurons in grafted mice at 10 mpt, showing a significant negative correlation.

## Data Availability

The data presented in this study are available in article.

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
