# Peer review of "Long-Term Evaluation of Intranigral Transplantation of Human iPSC-Derived Dopamine Neurons in a Parkinson’s Disease Mouse Model"

_cells, 2022, doi:10.3390/cells11101596_

Round 1

Reviewer 1 Report

The authors replied to the raised points accordingly.

I advise a final spelling check to identify few remaining typos (i.e specify line 768, HLA-matchning line 782)

Author Response

Thank you, we have checked the manuscript and  corrected the grammar mistakes as well as typos.

Reviewer 2 Report

Authors must include and discuss in the manuscript the articles published by Schweitzer et al (2020) and Takahashi et al (2020) regarding the iPSC-based clinical trials for PD.

Schweitzer, J.S., et al. 2020. Personalized iPSC-derived dopamine progenitor cells for Parkinson’s disease. N. Engl. J. Med. 382, 1926–1932.

Takahashi, J., 2020. iPS cell-based therapy for Parkinson’s disease: A Kyoto trial. Regen. Ther. 13, 18.

Author Response

Thank you, as suggester, we included and discussed in the manuscript the articles published by Schweitzer et al (2020) and Takahashi et al (2020) regarding the iPSC-based clinical trials for PD.

This manuscript is a resubmission of an earlier submission. The following is a list of the peer review reports and author responses from that submission.

Round 1

Reviewer 1 Report

The article from Brot and colleagues, shows a very interesting preclinical study consisting in a proof of principle of a cell therapy based on the transplantation of iPSC-derived dopamine neurons in the substantia nigra of an animal model of Parkinson’s disease. The study shows long-term survival of the transplanted dopamine progenitors/neurons and very interestingly a consistent reconstruction of the nigro-striatal pathway.

The paper is well written and scientifically comprehensive.

I have only few minor points to improve to final version of the manuscript:

1) In order to better characterize the behaviour of the of the transplanted progenitors it would be helpful to characterize how many of them are still in a proliferative state using a proliferation marker beside the nestin staining at different time points.

Author Response

We thank this Reviewer for his/her support and most useful criticisms, which we have tried to address in full, as detailed below.

To evaluate the proliferation of grafted cells, we performed a new series of experiments on brain sections at different times post transplantation: 1 mpt, 6 mpt, 8 mpt and 12 mpt using Ki-67 immunostaining and quantified the number Ki67+/HuNu+ cells within the grafts. We observed a significant decrease in cell proliferation rate over time (p=0.001). We found 9.78±0.96% of Ki-67+ cells at 1 mpt, 1.56±0.14% at 6 mpt, 0.27±0.03% at 8 mpt and 0.08±0.03% at 12 mpt.

We included these results in the manuscript new figure 8 (A-B) and new paragraph "3.6. Proliferation and apoptosis in grafts" and included some discussion of this aspect.

2) In a similar point of view it would be interesting also to have a characterization of apoptotic markers in the grafts at different time points.

Author Response

We completely agree with the Reviewer that that apoptosis is a crucial point for the long-term study of grafted cells. To address this point, we performed a new series of experiments on brain sections at different times post transplantation: 1 mpt, 6 mpt, 8 mpt and 12 mpt using an anti-Cleaved Caspase-3 antibody, a marker of programmed cell death [1]. We observed a significant increase in the number of Casp-3+/HuNu+ cells from 1 to 12 mpt (p=0.049). Indeed, we found 0.45±0.13% of apoptotic cells at 1 mpt, 0.75±0.14% at 6 mpt, 1.18±0.37% at 8 mpt and 1.35±0.16% at 12 mpt. Interestingly, apoptosis affects more specifically DA neurons, indeed, we observed a gradual increase in the number of TH+/Casp-3+/HuNu+ cell from 1 to 12 mpt. We found, 6.2±2.8% TH+/Casp-3+/HuNu+ cell, at 1 mpt, 19.6±2.6%, at 6 mpt, 39.1±4% at 8 mpt (p=0.045) and 44.2±4.3% at 12 mpt (p<0.011).

We included these results in the manuscript new figure 8 (C-D) and new paragraph "3.6. Proliferation and apoptosis in grafts".

In addition, we carried out the quantifications of the DA subtype at 6 mpt on 4 additional mice, to go from an initial n of 3, to n=7 now. “At 6 mpt, we observed 52.5±3.8% of TH/GIRK2+ neurons, and 39.9±2.2% of TH/Calbindin+ neurons. However, at 12 mpt, we observed a decrease in the percentage of TH/GIRK2+ neurons (p=0.0013) while the percentage of TH/Calbindin+ neurons remained identical to those observed at 6 mpt. Indeed, we observed 32.4±2% of TH/GIRK2+ neurons, and 41.8±1,4% of TH/Calbindin+ neurons.” These new data have been added to the graph of Figure 7C, and in the section "3.5 Cellular composition of the graft".

3) I think that the authors should also mention in the discussion recent clinical studies based on a similar approach (i.e. Gillmore et al., 2021, NEJM).

Author Response

We have included some discussion of this aspect.

References:

  1. Hernandez-Baltazar, D.; Mendoza-Garrido, M.E.; Martinez-Fong, D. Activation of GSK-3β and Caspase-3 Occurs in Nigral Dopamine Neurons during the Development of Apoptosis Activated by a Striatal Injection of 6-Hydroxydopamine. PLoS ONE 2013, 8, e70951, doi:10.1371/journal.pone.0070951.

Reviewer 2 Report

In this article, authors obtain midbrain dopaminergic neurons from iPSCs generated from human fibroblasts from a healthy donor and grafted these neurons into the SNpc in an animal model of Parkinson disease. Graft survival and maturation were analyzed from 1 to 12 months post-transplantation. Authors justify these studies to be used as a preclinical model before these cells can be used in future clinical trials.

However, there are other preclinical studies performed in different animal models including an in vivo study using monkey models as a simulation of a clinical trial. In this work, iPSCs were established from healthy and PD patients, and DA progenitor cells were transplanted into the bilateral putamen, and the monkeys were followed up for two years. The transplantation of both healthy individual- and patient- derived cells resulted in a significant behavioural improvement and dopamine synthesis by the transplanted cells.  Moreover, this preclinical study has been the basis for a clinical trial started in October 2018 at Kyoto University Hospital. In this trial, human iPSC-derived DA progenitor cells have been transplanted, with a surgical procedure, into the putamen of PD patients. Clinical trials of cell transplantation for PD using human ES cells are also currently underway in Australia, China and other new trials are about to start.

Author Response

We thank this Reviewer for his/her comments, however, we respectfully disagree with the reviewer's argument. Clinical trials with iPSC-derived DA neurons in PD patients are very promising, however, this approach is still in its nascent stages and several issues need to be addressed to effectively translate the iPSCs into the clinical setting without compromising patient safety. Indeed, there are many important questions still unanswered, such as the most appropriate developmental stage of DA neurons for transplantation, the extent to which in vitro-generated neurons correspond to in vivo- generated midbrain DA neurons, the site of transplantation, the long-term survival of transplanted neurons ...

One of the important messages of our study is the anatomical and functional reconstruction of midbrain degenerated pathways by intranigral transplantation of mDA neurons derived from hiPSCs. In the preclinical and clinical studies cited by the reviewer, the DA neurons were grafted in the striatum in an ectopic position. Currently a growing body of work is devoted to the benefit of intranigral transplantation of mDA neurons [1–4]. Recently, we performed a direct comparison of the potential benefits of intranigral versus intrastriatal grafts in animal models of Parkinson’s disease and reported that intranigral grafts promote better survival of dopaminergic neurons and that only intranigral grafts induced recovery of fine motor skills and normalized cortico- striatal responses [5]. All together, these studies highlight the importance of intranigral transplantation of mDA neurons derived hiPSC the importance of pursuing the work on the development of cell therapy approaches in Parkinson's disease.

References:

  1. Grealish, ; Diguet, E.; Kirkeby, A.; Mattsson, B.; Heuer, A.; Bramoulle, Y.; Van Camp, N.; Perrier, A.L.; Hantraye, P.; Björklund, A.; et al. Human ESC-Derived Dopamine Neurons Show Similar Preclinical Efficacy and Potency to Fetal Neurons When Grafted in a Rat Model of Parkinson’s Disease. Cell Stem Cell 2014, 15, 653– 665, doi:10.1016/j.stem.2014.09.017.
  2. Adler, F.; Cardoso, T.; Nolbrant, S.; Mattsson, B.; Hoban, D.B.; Jarl, U.; Wahlestedt, J.N.; Grealish, S.; Björklund, A.; Parmar, M. HESC-Derived Dopaminergic Transplants Integrate into Basal Ganglia Circuitry in a Preclinical Model of Parkinson’s Disease. Cell Reports 2019, 28, 3462-3473.e5, doi:10.1016/j.celrep.2019.08.058.
  3. Xiong, M.; Tao, Y.; Gao, Q.; Feng, B.; Yan, W.; Zhou, Y.; Kotsonis, T.A.; Yuan, T.; You, ; Wu, Z.; et al. Human Stem Cell-Derived Neurons Repair Circuits and Restore Neural Function. Cell Stem Cell 2021, 28, 112-126.e6, doi:10.1016/j.stem.2020.08.014.
  4. Moriarty, N.; Gantner, C.W.; Hunt, C.P.J.; Ermine, C.M.; Frausin, S.; Viventi, S.; Ovchinnikov, A.; Kirik, D.; Parish, C.L.; Thompson, L.H. A Combined Cell and Gene Therapy Approach for Homotopic Reconstruction of Midbrain Dopamine Pathways Using Human Pluripotent Stem Cells. Cell Stem Cell 2022, S1934590922000467, doi:10.1016/j.stem.2022.01.013.
  5. Droguerre, M.; Brot, S.; Vitrac, C.; Benoit-Marand, M.; Belnoue, L.; Patrigeon, M.; Lainé, A.; Béré, E.; Jaber, M.; Gaillard, A. Better Outcomes with Intranigral versus Intrastriatal Cell Transplantation: Relevance for Parkinson’s Disease. Cells 2022, 11, 1191, doi:10.3390/cells11071191.

Reviewer 3 Report

            In this manuscript, the authors perform a thorough characterization of (1) the differentiation of hiPSCs to dopaminergic neurons, and (2) the long-term survival and maturation of these cells upon transplantation into the mouse SNpc. Very few studies have performed as thorough an analysis on long term DA cell transplantations, so the authors should be commended for their work. Their findings underscore the notion that these types of long-term analysis are needed to fully understand the benefits and possible downside/side effects of hiPSC transplantations to treat neurological disease. I believe the authors can do a much better job in expanding their discussion to highlight many of the caveats and limitations of their study

  1. For Figures 1-4, the authors display immuostaining and quantification of numerous stem cell, neural progenitor, floorplate and DA markers. Because different markers are analyzed at different stages, it makes is somewhat challenging to follow various markers over time. I’d recommend the authors present quantification of the % cells expressing specific genes in a single line chart, with the X-axis representing timepoints analyzed (D21, D26, D36, D51). The Y-axis is the % of positive cells. Each gene would be a line on the graph displaying % cells over time, with each gene being a different color. This would allow the reader to compare all relevant genes in one figure, and should better emphasize changes over time (decrease of stem genes, increase of postmitotic and the DA specific genes, etc.).

Author Response

We thank this Reviewer for his/her support and most useful criticisms, which we have tried to address in full, as detailed below.

We have followed the suggestion of the reviewer, in order to have an overall view of the variation of each marker over time, we presented the % cells expressing specific genes in a single line, from Day 21 to Day 51. Indeed, this mode of presentation makes easier to follow the evolution of the different markers over time.

These data have been added to the graph of Figure 4C, and in the section "3.3. Maturation of DA neurons in vitro".

  1. One weakness of the manuscript is that the authors only used 1 hiPSC line for this study. Ideally researchers should use 2 different iPSC lines to ensure greater confidence that differentiation and transplantation results would be applicable to range of different iPSCs. It is not realistic to request that the author perform longterm transplantation analysis on another cell line for this study. However, the authors should mention this potential limitation in the discussion, noting that different iPSC lines display different differentiation efficiencies and likely response to transplants. Thus these approaches need to be replicated with additional cell lines in the future to instill greater confidence in the results.

Author Response

As rightfully pointed out by this reviewer, that different iPSC lines vary in their differentiation capacity and therefore therapeutic potential after transplantation. Each iPSC line needs to be pre-evaluated to ensure its validity and safety.

In a currently study, we generated several human iPSC lines from dermal fibroblasts of healthy and PD patient with different mutations. These iPSCs were differentiated into mDA neurons using the same protocol used in this manuscript. We performed an extensive in vitro and in vivo (post-transplant) analysis, similar to those performed in present manuscript. In vitro and in vivo analysis of iPSC show that they are similar in every aspect examined including pluripotency, differentiation into DA neurons, survival, and functional recovery. We are thus very confident about the robustness of the protocol used and the validity of the results obtained in our study results obtained by using hiPSC line mentioned in the manuscript.

In addition, we have included some discussion of this potential limitation.

Another issue the authors need to discuss in more detail is the potential benefits and drawbacks for using D26 as the transplantation date. The logic of why the authors chose this stage, rather than a later stage (D36 or maybe D51) when there was a significantly higher percentage of postmitotic neurons and DA-fated neurons, is unclear to me. Would the authors have still observed a relatively high percentage of immature cells (1/6th) 12 months after transplantation? Again, I don’t believe it’s fair to request the authors to repeat these long-term transplant experiments, but the authors must present a thorough analysis of these caveats in the Discussion. Maybe transplanting D36 cells would reduce/eliminate the number of immature neurons in the transplant and/or increase the % of DA neurons?

Author Response

We fully agree with the Reviewer that differentiation stage of the cells used for the transplantation impacts the efficiency of the transplantation.

Several groups focused on the identification of the most appropriate stage of development of DA neurons for transplantation. There is high variability in the results reported, the optimal differentiation stage of differentiation DA neurons for transplantation, can vary from Day 16 to Day 32 from one study to another [1]. However, the conclusion of these studies is that the transplantation of the cells starting from late midbrain floor-plate to an mDA neuroblast stage or early mDA neuron may be all suitable for robust survival and function.

Day 25 of differentiation corresponding to the stage of high NURR1 expression when most cells have started to exit cell cycle (early post-mitotic neurons) appear to produce rich mDA neuron grafts and have resulted in successful rescue of complex motor abnormalities in animal PD models including mouse, rat, and monkey [2,3].

For instance, Qiu et al., 2017, compared three differentiated stages of cells generated by a FP-based protocol: DA progenitors (D16), post-mitotic immature DA neurons (D25), and DA neurons (D35), and examined the competence of these cells in cell- transplantation therapy in a mouse model of PD [4]. They identified post-mitotic immature DA neurons (D25, high NURR1 expression) as the most suitable cell source for transplantation. Other teams such as the group by Takahashi, transplanted at day 28 mDA “progenitors,” which are differentiated further starting from floor-plate progenitor stage sorted by CORIN at day 12 [5,6]. Parmar group demonstrated that relatively early stage, day 16 mDA progenitors, just beyond floor-plate precursor stage are also suitable for intracerebral transplantations [7–12].

The conclusion of these studies is that the transplantation of the cells starting from late midbrain floor-plate to an mDA neuroblast stage or early mDA neuron may be suitable for robust survival and function of grafted neurons. We have included some discussion of this aspect.

Minor Comments

  1. The authors should cite a very recent Cell Stem Cell paper that also transplanted hiPSC-derived DA neurons into the SNpc and performed similar types of long term analysis (albeit not up to 1 year): Moriarty…Thompson Cell Stem Cell 2022.

Author Response

We have further discussed our results, including citations of recent clinical and pre- clinical studies [13-21].

  1. Fig 2A, the authors should write the days in culture in the cell images, it’s not clear to me what day the images are from.

Author Response

This is now corrected in Figure 2A, culture days have been added.

  1. Fig 3A-B. Lmx1a decreased significantly more at D26 compared to the other FP markers (OTX2 & FOXA2). Based on the known expression patterns of these genes, was this decrease expected? Does this reduction have biological implications?

Author Response

Sustained expression of Lmx1a and Lmx1b is required for the survival of adult midbrain dopaminergic neurons [22]. The published results concerning the expression of LMX1A are inconsistent. It has been reported that expression of LMX1a in post-mitotic neurons decreases with aging [23]. However, other recent studies have shown that the expression level of LMX1a is maintained in vitro for more than 40 days [6,24]. Concerning our results, it is true that the expression level of LMX1a is lower than OTX2 & FOXA2 but at D36 the expression level of LMX1a is the same as FOXA2.

References:

  1. Kim, T.W.; Koo, S.Y.; Studer, L. Pluripotent Stem Cell Therapies for Parkinson Disease: Present Challenges and Future Opportunities. Front Cell Dev Biol 2020, 8, 729, doi:10.3389/fcell.2020.00729.
  2. Kriks, S.; Shim, J.-W.; Piao, J.; Ganat, Y.M.; Wakeman, D.R.; Xie, Z.; Carrillo-Reid, ; Auyeung, G.; Antonacci, C.; Buch, A.; et al. Dopamine Neurons Derived from Human ES Cells Efficiently Engraft in Animal Models of Parkinson’s Disease. Nature 2011, 480, 547–551, doi:10.1038/nature10648.
  3. Ganat, Y.M.; Calder, E.L.; Kriks, S.; Nelander, J.; Tu, E.Y.; Jia, F.; Battista, D.; Harrison, ; Parmar, M.; Tomishima, M.J.; et al. Identification of Embryonic Stem Cell– Derived Midbrain Dopaminergic Neurons for Engraftment. J. Clin. Invest. 2012, 122, 2928–2939, doi:10.1172/JCI58767.
  4. Steinbeck, J.A.; Choi, S.J.; Mrejeru, A.; Ganat, Y.; Deisseroth, K.; Sulzer, D.; Mosharov, V.; Studer, L. Optogenetics Enables Functional Analysis of Human Embryonic Stem Cell-Derived Grafts in a Parkinson’s Disease Model. Nat Biotechnol 2015, 33, 204–209, doi:10.1038/nbt.3124.
  5. Qiu, ; Liao, M.-C.; Chen, A.K.; Wei, S.; Xie, S.; Reuveny, S.; Zhou, Z.D.; Hunziker, W.; Tan, E.K.; Oh, S.K.W.; et al. Immature Midbrain Dopaminergic Neurons Derived from Floor-Plate  Method       Improve Cell Transplantation    Therapy      Efficacy      for Parkinson’s   Disease. Stem    Cells           Transl         Med      2017,    6,           1803–1814, doi:10.1002/sctm.16-0470.
  6. Doi, D.; Samata, B.; Katsukawa, M.; Kikuchi, T.; Morizane, A.; Ono, Y.; Sekiguchi, ; Nakagawa, M.; Parmar, M.; Takahashi, J. Isolation of Human Induced Pluripotent Stem Cell-Derived Dopaminergic Progenitors by Cell Sorting for Successful Transplantation. Stem Cell Reports 2014, 2, 337–350, doi:10.1016/j.stemcr.2014.01.013.
  7. Kikuchi, ; Morizane, A.; Doi, D.; Magotani, H.; Onoe, H.; Hayashi, T.; Mizuma, H.; Takara, S.; Takahashi, R.; Inoue, H.; et al. Human IPS Cell-Derived Dopaminergic Neurons Function in a Primate Parkinson’s Disease Model. Nature 2017, 548, 592– 596, doi:10.1038/nature23664.
  8. Grealish, ; Diguet, E.; Kirkeby, A.; Mattsson, B.; Heuer, A.; Bramoulle, Y.; Van Camp, N.; Perrier, A.L.; Hantraye, P.; Björklund, A.; et al. Human ESC-Derived Dopamine Neurons Show Similar Preclinical Efficacy and Potency to Fetal Neurons When Grafted in a Rat Model of Parkinson’s Disease. Cell Stem Cell 2014, 15, 653– 665, doi:10.1016/j.stem.2014.09.017.
  9. Adler, F.; Cardoso, T.; Nolbrant, S.; Mattsson, B.; Hoban, D.B.; Jarl, U.; Wahlestedt, J.N.; Grealish, S.; Björklund, A.; Parmar, M. HESC-Derived Dopaminergic Transplants Integrate into Basal Ganglia Circuitry in a Preclinical Model of Parkinson’s Disease. Cell Reports 2019, 28, 3462-3473.e5, doi:10.1016/j.celrep.2019.08.058.
  10. Kirkeby, ; Grealish, S.; Wolf, D.A.; Nelander, J.; Wood, J.; Lundblad, M.; Lindvall, O.; Parmar, M. Generation of Regionally Specified Neural Progenitors and Functional Neurons from Human Embryonic Stem Cells under Defined Conditions. Cell Reports 2012, 1, 703–714, doi:10.1016/j.celrep.2012.04.009.
  11. Kirkeby, A.; Nolbrant, S.; Tiklova, K.; Heuer, A.; Kee, N.; Cardoso, T.; Ottosson, R.; Lelos, M.J.; Rifes, P.; Dunnett, S.B.; et al. Predictive Markers Guide Differentiation to Improve Graft Outcome in Clinical Translation of HESC-Based Therapy for Parkinson’s Disease. Cell Stem Cell 2017, 20, 135–148, doi:10.1016/j.stem.2016.09.004.
  12. Grealish, S.; Heuer, A.; Cardoso, T.; Kirkeby, A.; Jönsson, M.; Johansson, J.; Björklund, ; Jakobsson, J.; Parmar, M. Monosynaptic Tracing Using Modified Rabies Virus Reveals Early and Extensive Circuit Integration of Human Embryonic Stem Cell- Derived Neurons. Stem Cell Reports 2015, 4, 975–983, doi:10.1016/j.stemcr.2015.04.011.
  13. Cardoso, ; Adler, A.F.; Mattsson, B.; Hoban, D.B.; Nolbrant, S.; Wahlestedt, J.N.; Kirkeby, A.; Grealish, S.; Björklund, A.; Parmar, M. Target-Specific Forebrain Projections and Appropriate Synaptic Inputs of HESC-Derived Dopamine Neurons Grafted to the Midbrain of Parkinsonian Rats. J Comp Neurol 2018, 526, 2133–2146, doi:10.1002/cne.24500.
  14. Parmar, M.; Grealish, S.; Henchcliffe, C. The Future of Stem Cell Therapies for Parkinson Disease. Nat Rev Neurosci 2020, 21, 103–115, doi:10.1038/s41583-019- 0257-7.
  15. Takahashi, Preclinical Evaluation of Patient-Derived Cells Shows Promise for Parkinson’s Disease.   Journal   of   Clinical   Investigation   2020,   130, 601–603, doi:10.1172/JCI134031.
  16. Barker, R.A.; TRANSEURO consortium Designing Stem-Cell-Based Dopamine Cell Replacement Trials for Parkinson’s Disease. Nat Med 2019, 25, 1045–1053, doi:10.1038/s41591-019-0507-2.
  17. Gillmore, J.D.; Gane, E.; Taubel, J.; Kao, J.; Fontana, M.; Maitland, M.L.; Seitzer, ; O’Connell, D.; Walsh, K.R.; Wood, K.; et al. CRISPR-Cas9 In Vivo Gene Editing for Transthyretin Amyloidosis. N Engl J Med 2021, 385, 493–502, doi:10.1056/NEJMoa2107454.
  18. Gordián-Vélez, W.J.; Browne, K.D.; Galarraga, J.H.; Duda, J.E.; España, R.A.; Chen, H.I.; Burdick, J.A.; Cullen, D.K. Dopaminergic Axon Tracts within a Hyaluronic Acid Hydrogel Encasement for Implantation to Restore the Nigrostriatal Pathway. bioRxiv 2021, 07.03.451006, doi:10.1101/2021.07.03.451006.
  19. Aldrin-Kirk, ; Åkerblom, M.; Cardoso, T.; Nolbrant, S.; Adler, A.F.; Liu, X.; Heuer, A.; Davidsson, M.; Parmar, M.; Björklund, T. A Novel Two-Factor Monosynaptic TRIO Tracing Method for Assessment of Circuit Integration of HESC-Derived Dopamine Transplants. Stem Cell Reports 2021, S2213-6711(21)00595-6, doi:10.1016/j.stemcr.2021.11.014.
  20. Hunt, P.J.; Penna, V.; Gantner, C.W.; Moriarty, N.; Wang, Y.; Franks, S.; Ermine, C.M.; Luzy, I.R.; Pavan, C.; Long, B.M.; et al. Tissue Programmed Hydrogels Functionalized with GDNF Improve Human Neural Grafts in Parkinson’s Disease. Adv. Funct. Mater. 2021, 31, 2105301, doi:10.1002/adfm.202105301.
  21. Moriarty, N.; Gantner, C.W.; Hunt, C.P.J.; Ermine, C.M.; Frausin, S.; Viventi, S.; Ovchinnikov, A.; Kirik, D.; Parish, C.L.; Thompson, L.H. A Combined Cell and Gene Therapy Approach for Homotopic Reconstruction of Midbrain Dopamine Pathways Using Human Pluripotent Stem Cells. Cell Stem Cell 2022, S1934590922000467, doi:10.1016/j.stem.2022.01.013.
  22. Doucet-Beaupré, ; Gilbert, C.; Profes, M.S.; Chabrat, A.; Pacelli, C.; Giguère, N.; Rioux, V.; Charest, J.; Deng, Q.; Laguna, A.; et al. Lmx1a and Lmx1b Regulate Mitochondrial Functions and Survival of Adult Midbrain Dopaminergic Neurons. Proc Natl Acad Sci U S A 2016, 113, E4387-4396, doi:10.1073/pnas.1520387113.
  23. Doucet-Beaupré, H.; Ang, S.-L.; Lévesque, M. Cell Fate Determination, Neuronal Maintenance and Disease State: The Emerging Role of Transcription Factors Lmx1a and FEBS Letters 2015, 589, 3727–3738, doi:10.1016/j.febslet.2015.10.020.
  24. Doi, D.; Magotani, H.; Kikuchi, T.; Ikeda, M.; Hiramatsu, S.; Yoshida, K.; Amano, ; Nomura, M.; Umekage, M.; Morizane, A.; et al. Pre-Clinical Study of Induced Pluripotent Stem Cell-Derived Dopaminergic Progenitor Cells for Parkinson’s Disease. Nat Commun 2020, 11, 3369, doi:10.1038/s41467-020-17165-w.